# PUBDEF: DEFENDING AGAINST TRANSFER ATTACKS FROM PUBLIC MODELS

**Chawin Sitawarin**
UC Berkeley

**Jaewon Chang**[*]
UC Berkeley

**David Huang**[*]
UC Berkeley

**Wesson Altoyan**
King Abdulaziz City for Science and Technology

**David Wagner**
UC Berkeley

## ABSTRACT

Adversarial attacks have been a looming and unaddressed threat in the industry. However, through a decade-long history of the robustness evaluation literature, we have learned that mounting a strong or optimal attack is challenging. It requires both machine learning and domain expertise. In other words, the white-box threat model, religiously assumed by a large majority of the past literature, is unrealistic. In this paper, we propose a new practical threat model where the adversary relies on **transfer attacks through publicly available surrogate models**. We argue that this setting will become the most prevalent for security-sensitive applications in the future. We evaluate the transfer attacks in this setting and propose a specialized defense method based on a game-theoretic perspective. The defenses are evaluated under 24 public models and 11 attack algorithms across three datasets (CIFAR-10, CIFAR-100, and ImageNet). Under this threat model, our defense, PUBDEF, outperforms the state-of-the-art white-box adversarial training by a large margin with **almost no loss in the normal accuracy**. For instance, on ImageNet, our defense achieves 62% accuracy under the strongest transfer attack vs only 36% of the best adversarially trained model. Its accuracy when not under attack is only 2% lower than that of an undefended model (78% vs 80%). Code is available here.

## 1 INTRODUCTION

Current ML models are fragile: they are susceptible to adversarial examples (Biggio et al., 2013; Szegedy et al., 2014; Goodfellow et al., 2015), where a small imperceptible change to an image radically changes its classification. This has stimulated a profusion of research on ML-based methods to improve the robustness to adversarial attacks. Unfortunately, progress has slowed and fallen far short of what is needed to protect systems in practice (Hendrycks et al., 2022). In this paper, we articulate a new approach that we hope will lead to pragmatic improvements in resistance to attacks.

In the literature, we can find two defense strategies: systems-level defenses and ML-level defenses. Systems-level defenses include controls such as keeping the model weights secret, returning only the predicted class and not confidence scores, monitoring use of the model to detect anomalous patterns, etc. Unfortunately, systems-level defenses have proven inadequate on their own: for instance, transfer attacks can successfully attack a target model even without knowing its weights. Therefore, most research focuses on ML-level defenses, where we try to build models that are more robust against such attacks, for example through novel architectures and/or training methods. Researchers made early progress on ML-level defenses, with the introduction of adversarial training (Madry et al., 2018), but since then progress has slowed dramatically, and there is no clear path to achieving strong adversarial robustness in any deployable model.

Current ML-level defenses suffer from two major problems: first, they have an unacceptable negative impact on clean accuracy (accuracy when not under attack), and second, they focus on a threat model that is increasingly recognized to be unrealistic (Gilmer et al., 2018; Goodfellow, 2018; Hendrycks et al., 2022; Apruzzese et al., 2023). These problems are closely related: prevailing academic threat models are unrealistic as they grant the attacker excessive powers that are hard to realize in real life,

---

[*]Equal contribution. Corresponding email: `chawins@berkeley.edu`

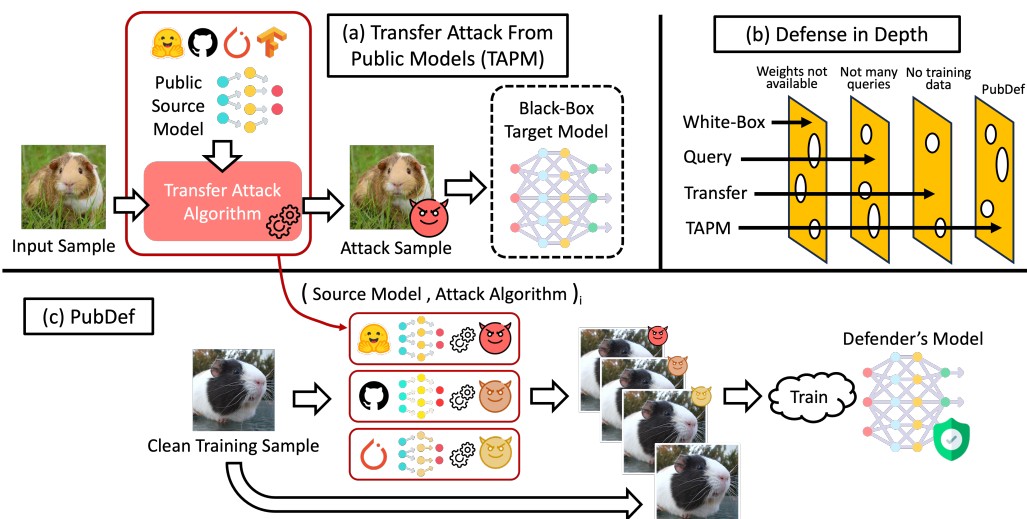

Figure 1: (a) Proposed threat model: transfer attack with public source models (TAPM). We consider a low-cost black-box adversary who generates adversarial examples from publicly available models with a known attack algorithm. (b) Our approach is based on stopping each major category of attack with a combination of multiple mechanisms. (c) Our defense, PUBDEF, trains the defended model to resist transfer attacks from several publicly available source models. Our model is robust to a wide range of transfer attacks, including both those from source models that were trained against and others that were not trained against, while also maintaining high clean accuracy.

making it too difficult to achieve strong security against such unrealistic attacks. Because of the existing trade-off between adversarial robustness and clean accuracy, this in turn means achieving any non-trivial adversarial robustness currently requires unacceptable degradation to clean accuracy.

We advocate for a different approach, captured in slogan form as follows:

> Secure ML   =   Realistic threat model   +   Systems-level defenses
> +   ML-level defenses against those threats

We propose using all available systems-level defenses. We articulate a concrete threat model, influenced by what attacks cannot be stopped by systems-level defenses. Specifically, we propose *security against transfer attacks from public models* (TAPM; Fig. 1(a)) as the threat model we focus on. The TAPM threat model focuses on transfer attacks where the adversary starts with a publicly available model, attacks the public model, and then hopes that this attack will "transfer", i.e., will also be effective against the target model. Because public models are often widely available, e.g., in model zoos, this kind of transfer attack is particularly easy to mount and thus particularly important to defend against. Under the TAPM threat model, we assume neither the model weights nor training set are known to the attacker, and the attacker cannot train their own model or mount query-based attacks that involve querying the target model many times. These assumptions are driven partly by what kinds of attacks can be prevented or mitigated by existing systems-level defenses.

Finally, we introduce PUBDEF (Fig. 1(c)), a new method for training models that will be secure against transfer attacks from public models. PUBDEF models are attractive for practical deployment. For instance, they achieve clean accuracy close to that of an undefended model, so there is little loss in performance when not under attack. When under attack (via transfer from public models), adversarial accuracy remains fairly high: 88.6% for CIFAR-10 (almost 20 points higher than any previous defense), 50.8% for CIFAR-100 (18 points higher), and 62.3% for ImageNet (26 points higher than any previous defense). While our defense is not perfect and is not appropriate in all scenarios, we believe it is a pragmatic defense that can be deployed without major loss of clean accuracy, while making life as difficult for attackers as possible within that constraint.

## 2    RELATED WORK

We provide an introduction to several types of attacks and threat models seen in the literature, for comparison to our new threat model.

| Defenses | CIFAR-10 | | CIFAR-100 | | ImageNet | |
|---|---|---|---|---|---|---|
| | Clean | Adv. | Clean | Adv. | Clean | Adv. |
| No defense | 96.3 | 0.0 | 81.5 | 0.0 | 80.4 | 0.0 |
| Best white-box adv. train | 85.3 | 68.8 | 68.8 | 32.8 | 63.0 | 36.2 |
| DVERGE + adv. train | 87.6 | 59.6 | 6.3 | 2.1[*] | | |
| TRS + adv. train | 86.9 | 66.7 | 63.9 | 39.1 | | |
| **PUBDEF (ours)** | 96.1 (+10.8) | 88.6 (+19.8) | 76.2 (+7.4) | 50.8 (+18.0) | 78.6 (+15.6) | 63.0 (+26.8) |

Table 1: Clean and adversarial accuracy of PUBDEF vs the best previously published defenses against transfer attacks. Adversarial accuracy is measured in the TAPM threat model. "White-box adv. train" are the most robust models from ROBUSTBENCH which share the same architecture as PUBDEF. DVERGE (Yang et al., 2020) and TRS (Yang et al., 2021) are two state-of-the-art defenses against transfer attacks. [*]DVERGE is designed for CIFAR-10 and is difficult to train on the other datasets. TRS/DVERGE with adversarial training is not included for ImageNet due to its computation cost.

**White-box attacks.** In this threat model, the attacker is assumed to know everything about the target model, including all model weights. This is the most studied threat model in the literature. Adversarial training (Madry et al., 2018) has been the primary defense against white-box adversarial examples. However, adversarial training sacrifices a considerable amount of clean accuracy (Tsipras et al., 2019), rendering it unattractive to deploy in practice.

**Transfer attacks.** Papernot et al. (2016) first demonstrated the threat of transfer attacks: adversarial examples generated on one ML model (the surrogate) can successfully fool another model if both models are trained on the same task. Liu et al. (2017); Tramèr et al. (2017); Demontis et al. (2019) propose various methods for quantifying a degree of the attack transferability including the distance to the decision boundary as well as the angle between the gradients of two models. A great number of transfer attack algorithms have been proposed over the years (Zhao et al., 2022), e.g., using momentum during optimization (Dong et al., 2018; Lin et al., 2020; Wang et al., 2021b), applying data augmentation (Xie et al., 2019; Wang et al., 2021a; Lin et al., 2020), and alternative loss functions (Zhang et al., 2022; Huang et al., 2019). In this threat model, researchers often assume that the training set for the defended model is available to the attacker, and the attacker can either train their own surrogate models or use publicly available models as a surrogate.

**Query-based attacks.** Consider the setting where the attacker can query the target model (submit an input and obtain the classifier's output for this input), but does not know the model's weights. This threat model is particularly relevant for classifiers made available via an API or cloud service. Attacks can iteratively make a series of queries to learn the decision boundary of the model and construct an adversarial example (Brendel et al., 2018; Ilyas et al., 2018; Andriushchenko et al., 2020). Systems-level defenses include returning only hard-label predictions (the top-1 predicted class but not the confidence level), rate-limiting, and monitoring queries to detect query-based attacks (Biggio et al., 2013; Goodfellow, 2019; Chen et al., 2020b; Li et al., 2020).

## 3 THREAT MODEL

We first define our threat model for this paper: *transfer attack with public models* (TAPM). It is designed to capture a class of attacks that are especially easy and low-cost to mount, do not require great sophistication, and are not easily prevented by existing defenses. It fills in a part of the attack landscape that has not been captured by other well-known threat models (e.g., white-box, query-based, and transfer attack). Under TAPM, the adversary has the following capabilities:

1. They have white-box access to all publicly available models trained for the same task. They can mount a transfer attack, using any public model as the surrogate.

2. They cannot train or fine-tune a neural network. This might be because a reasonable training set is not publicly available, or because the training process requires substantial expertise and resources that outweighs the economic gain of the attack.

3. They can submit one or more adversarial inputs to the target model but cannot run query-based attacks. This assumption is particularly well-suited to security-sensitive tasks, e.g., authentication and malware detection, where the adversary is caught immediately if the attack fails, or to systems where other effective defenses against query-based attacks can be deployed.

By default, we assume that the defender is also aware of the same set of public models. However, we will later show that our defense generalizes exceptionally well to unseen public models.

**Notation.** Let $\mathcal{S} = \{S_1, \ldots, S_s\}$ denote a finite set of all public models on the same task and $\mathcal{A} = \{A_1, \ldots, A_a\}$ a set of known transfer attack algorithms. An attack generates an $\ell_p$-norm bounded adversarial example from a model A and an input sample $x$:

$$x_{\text{adv}} = A(S, (x, y)) \quad \text{such that} \quad \|x_{\text{adv}} - x\|_p \leq \epsilon \tag{1}$$

where $S \in \mathcal{S}$ is a source or surrogate model. A transfer attack $x_{\text{adv}}$ is uniquely defined by a pair $(S, A)$. The attack is then evaluated on a target model $T \notin \mathcal{S}$ and considered successful if $T(x_{\text{adv}}) \neq y$.

## 4 GAME-THEORETIC PERSPECTIVE

We begin by motivating our defense through a game-theoretic lens. Prior work has formulated adversarial robustness as a two-player zero-sum game (Araujo et al., 2020; Meunier et al., 2021; Rathbun et al., 2022) but under different threat models and contexts. Under our TAPM setup, the attacker's strategy is naturally discrete and finite. The attacker chooses a source model $S \in \mathcal{S}$ and an attack algorithm $A \in \mathcal{A}$ and obtains an adversarial sample $x_{\text{adv}}$ (as defined in Eq. (1)). Essentially, each pair $(S, A)$ corresponds to one of $|\mathcal{S}| \cdot |\mathcal{A}| = s \cdot a$ attack strategies. We will describe two versions of the game with different defender strategies.

### 4.1 SIMPLE GAME

As a warm-up, we will first consider a discrete defense strategy where the defender trains $s \cdot a$ models, one against each of the attack strategies. Denote a defender's model by $T \in \mathcal{T}$ where $|\mathcal{T}| = s \cdot a$. The defender's strategy is to choose $T$ to classify a given $x_{\text{adv}}$ where $T$ is trained to minimize the expected risk of both the normal samples and the transfer adversarial samples $x_{\text{adv}}$ from Eq. (1).

$$\underset{\theta}{\arg\min} \; \mathbb{E}_{x,y} \left[ L(T_\theta(x), y) + L(T_\theta(x_{\text{adv}}), y) \right] \tag{2}$$

Note that this formulation is similar to the well-known adversarial training (Goodfellow et al., 2015; Madry et al., 2018) except that $x_{\text{adv}}$ is independent to $\theta$ or the model being trained. The payoff of the defender is defined as *the expected accuracy* on $x_{\text{adv}}$ chosen by the attacker:

$$r_D(\boldsymbol{\pi}_A, \boldsymbol{\pi}_D) = \mathbb{E}_{T \sim \boldsymbol{\pi}_D} \mathbb{E}_{(S,A) \sim \boldsymbol{\pi}_A} \mathbb{E}_{x,y} [\mathbb{1}\{T(A(S, (x, y))) = y\}] \tag{3}$$

where $\boldsymbol{\pi}_A, \boldsymbol{\pi}_D$ are *mixed* (i.e., potentially randomized) strategies for the attacker and the defender, respectively. In other words, $\boldsymbol{\pi}_A, \boldsymbol{\pi}_D$ each represents a multinomial distribution over the $s \cdot a$ pure (i.e., non-randomized) strategies. The attacker's payoff is $r_A(\boldsymbol{\pi}_A, \boldsymbol{\pi}_D) = -r_D(\boldsymbol{\pi}_A, \boldsymbol{\pi}_D)$. The payoff matrix $\boldsymbol{R} \in \mathbb{R}^{sa \times sa}$ is defined by $\boldsymbol{R}_{i,j} = r_D(\boldsymbol{e}_i, \boldsymbol{e}_j)$.

As an example, we empirically compute $\boldsymbol{R}$ (Fig. 2) choosing $\mathcal{S} = \{S_1, \ldots, S_4\}$ to be four public models and $\mathcal{A}$ as only PGD attack (Madry et al., 2018). We will later describe how these models are chosen in Section 5.2. The defender also has four models $\mathcal{T} = \{T_1, \ldots, T_4\}$, where $T_i$ is adversarially trained to be robust against transfer attacks from $S_i$. Notice that the diagonal entries are large because $T_i$ is trained on the attack from $S_i$. Von Neumann's minimax theorem guarantees the existence of a Nash equilibrium, i.e., an optimal strategy for each player (v. Neumann, 1928) (see Appendix A.3). The optimal strategy can be efficiently computed using linear programming (van den Brand, 2020). For the payoff

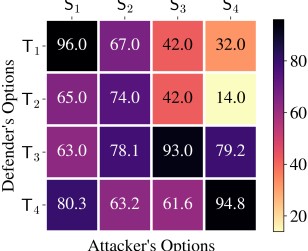

Figure 2: The payoff matrix of the simple game.

matrix in Fig. 2, the expected payoff for the optimal strategy is 73.0, meaning that when both the attacker and the defender choose their strategies optimally, the target model can achieve 73.0% accuracy on average. This is reasonable, but as we show next, we can do better.

### 4.2 COMPLEX GAME

Now we make the defender's action space more flexible: the defender can choose their model's weights arbitrarily, instead of being limited to one of $s \cdot a$ models. We extend the loss function in Eq. (2) to represent the loss against a transfer attack chosen according to mixed strategy $\pi_A$:

$$\underset{\theta}{\arg\min} \; \mathbb{E}_{x,y} \left[ L(f_\theta(x), y) + \sum_{i=1}^{s \cdot a} \pi_i L\left(f_\theta\left(x_{\text{adv},i}\right), y\right) \right] \tag{4}$$

where $x_{\mathrm{adv},i}$ is an attack generated by the $i$-th attack strategy, and the attacker's (mixed) strategy is given by $\boldsymbol{\pi} = (\pi_1, \ldots, \pi_{sa})$ representing a probability distribution over the $s \cdot a$ (pure) attack strategies. Note that we can recover the simple game if the adversary is restricted to choosing $\pi_i$'s s.t. $\pi_i = 1$ for a single $i$ and 0 otherwise. However, when $\boldsymbol{\pi}$ represents any probability distribution, the reward function is no longer linear in $\boldsymbol{\pi}$ so von Neumann's minimax theorem no longer applies. A Nash equilibrium may exist, but there is no known efficient algorithm to compute it.[1]

One naive strategy for the defender is to assume the attacker will choose uniformly at random from all $s \cdot a$ attacks and find the best response, i.e., find model weights $\theta$ that minimize Eq. (4) when $\pi_i = 1/sa$ for all $i$'s. This amounts to adversarially training a model against this particular (mixed) attacker strategy. In this case, the defender's payoff (adversarial accuracy) against each of the four attacks turns out to be $[96.3, 90.4, 94.6, 96.0]$, for the payoff matrix in Fig. 2. This means the defender achieves over 90% accuracy against all four transfer attacks, which is a significant improvement over the equilibrium of the simple game (73%). This suggests that while we may not be able to solve the complex game optimally, it already enables a much better strategy for the defender. In Section 5, we will explore several heuristics to approximately find a "local" equilibrium.

## 5 OUR PRACTICAL DEFENSE

### 5.1 LOSS FUNCTION AND WEIGHTING CONSTANTS

We propose several heuristics for solving the complex game (Section 4.2) that do not require explicitly computing the full payoff matrix. Instead, the defender trains only one model with adjustable weights $\pi_i$'s as defined in Eq. (4). We experiment with the three training losses (detailed in Appendix A.4) and report the best one. Overall, we find that simply sampling one attack randomly (a pair of $(\mathsf{S}, \mathsf{A})$) at each iteration performs well or best in most cases.

### 5.2 DEFENDER'S SOURCE MODEL SELECTION

Given that the set of publicly available models is public and known to both the attacker and the defender, the most natural choice for the defender is to train against *all* publicly available models. However, the computation cost can be prohibitive. We show that we can achieve nearly as good performance by choosing only a small subset of the publicly available models. However, finding an optimal set of source models is non-trivial without trying out all possible combinations.

Intuitively, to be robust against a wide range of transfer attacks, the defender should train the target model against a diverse set of source models and algorithms. The "diversity" of a set of models is challenging to define. Natural approaches include using a diverse set of architectures (e.g., ConvNet vs Transformer), a diverse set of (pre-)training methodologies (e.g., supervised vs unsupervised), and a diverse set of data augmentation strategies. In our experiments, we found that the *training procedure*—namely (1) normal, (2) $\ell_\infty$-adversarial, (3) $\ell_2$-adversarial, or (4) corruption-robust training—has the largest effect on the defense. For the rest of the paper, we categorize the source models into one of these four groups.[2] We will discuss the implications of this grouping in Sections 6.2 and 7.1.

This motivates a simple yet surprisingly effective heuristic that we use for selecting the set of source models: when training PUBDEF, we use one source model from each group (four source models in total for CIFAR-10 and three for CIFAR-100 and ImageNet). In more detail, we first choose four source models: the public model that is most robust against $\ell_\infty$ white-box attacks, the public model that is most robust against $\ell_2$ white-box attacks, the public model that is most corruption robust, and one arbitrary public model that is normally trained. Then, we compute the adversarial accuracy against transfer attacks from every publicly available model. If the adversarial accuracy against transfer attacks from some other public model $\mathsf{S}'$ is significantly lower than the adversarial accuracy against transfer attacks from $\mathsf{S}$ (the chosen model in the same group as $\mathsf{S}'$), then we swap in $\mathsf{S}'$ and remove $\mathsf{S}$. We made one swap for CIFAR-100 and ImageNet and no swap for CIFAR-10. We find that this simple heuristic works well in practice and performs better than a random subset (Section 7.1).

---

[1]If we discretize each $\pi_i$, the equilibrium is still guaranteed to exist by Nash's theorem (Nash, 1951), but we need to deal with an exponential (in $sa$) number of defense models.

[2]For CIFAR-100 and ImageNet, it is hard to find $\ell_2$-adversarially trained models that are publicly available, so we exclude this group and only consider the remaining three (normal, $\ell_\infty$, and corruption).

# 6 EXPERIMENTS

## 6.1 SETUP

**Metrics.** The two most common metrics used to compare models in the past literature are clean and adversarial accuracy. The clean accuracy is simply the accuracy on the test set, with no attack. There are multiple ways to measure the adversarial accuracy under our threat model depending on the attacker's strategy (e.g., average or worst-case). We conservatively assume that the attacker knows the defender's strategy and chooses the best pair of $(S, A)$ to run the attack.[3] In other words, we report the adversarial accuracy against the worst-case TAPM attack.

**Baseline defenses.** Adversarial training (Madry et al., 2018) can defend against transfer attacks but at the cost of excessive degradation to clean accuracy (as we will show next). We compare PUBDEF to the best white-box adversarially trained (AT) model from ROBUSTBENCH that has the same architecture. For CIFAR-10, CIFAR-100, and ImageNet, the best are Addepalli et al. (2022b), Addepalli et al. (2022a), and Salman et al. (2020), respectively. Additionally, we compare our method to DVERGE (Yang et al., 2020) and TRS (Yang et al., 2021), the two state-of-the-art defenses against transfer attacks. These use an ensemble of models for greater diversity, so that an attack that succeeds on one might fail on another, hopefully making transfer attacks harder.

**Public source models.** For each dataset, we select 24 public pre-trained models: 12 normally trained and 12 robustly trained models. The normal group comes from multiple public model zoos including Hugging Face (Face, 2023), `timm` (Wightman, 2019), and (for CIFAR-10/100) two Github repositories (Chen, 2023; Phan, 2023). The robust models are all hosted on ROBUSTBENCH (Croce et al., 2020). We select a wide variety of models based on their architecture (e.g., assorted ConvNet, ViT (Dosovitskiy et al., 2021), Swin (Liu et al., 2021), BeiT (Bao et al., 2022), ConvMixer (Trockman & Kolter, 2023), zero-shot CLIP (Radford et al., 2021), etc.) and training methods to ensure high diversity. We try our best to not select two models with the same architecture or from the same paper.

**Attack algorithms.** We evaluate the robustness of the defenses with 11 different attack algorithms. All of the attacks are gradient-based, but they utilize different techniques for improving the transferability of the attacks, e.g., momentum, data augmentation, intermediate representation, etc. Please refer to Appendix A.1 for the complete list of both the source models and the attack algorithms.

**PUBDEF training.** We adversarially train the defended model by selecting a subset of source models according to the heuristic in Section 5.2 and then using a PGD transfer attack on that subset. We use a WideResNet-34-10 architecture for CIFAR-10/100 and ResNet-50 for ImageNet. We do not use any extra data or generated data for training. Appendix A.1.3 has the complete training details.

## 6.2 RESULTS

**PUBDEF is more robust to all 264 transfer attacks than the baselines by a large margin.** We generate 264 transfer attacks, one for each of 11 attack algorithms and 24 source models, and evaluate PUBDEF according to the most successful attack. For CIFAR-10, PUBDEF achieves 20 percentage points higher adversarial accuracy than the best previously published defense, with comparable clean accuracy to an undefended model (Table 1). For CIFAR-100, PUBDEF achieves 10–18 p.p. higher robustness and 7–12 p.p. higher clean accuracy than the best prior defense. For ImageNet, PUBDEF achieves 26 p.p. better adversarial accuracy and 15 p.p. better clean accuracy than adversarial training; its clean accuracy is only 2 p.p. lower than an undefended model. It is beyond our resources to train TRS and DVERGE for ImageNet, due to the combination of ensembling and adversarial training.

Fig. 3 shows all adversarial accuracies of PUBDEF by pairs of $(S, A)$ on ImageNet. Here, the overall best attack is NI-Admix-TI-DIM from a ResNet-50 on Hugging Face. M-PGD, Admix, and NI-Admix-TI-DIM are generally the strongest attack algorithms across source models and datasets. Surprisingly, PGD usually performs well above average compared to the other more sophisticated attacks. Appendix A.6.2 shows more detail.

**PUBDEF maintains high clean accuracy.** Compared to the state-of-the-art top-1 clean accuracy for undefended models, our defense experiences only a 0–5% drop in the clean accuracy (Table 1). Compare to white-box adversarial training, which suffers a 11–18% drop. We emphasize that the

---

[3]This assumption is realistic because, with limited queries, the attacker can intelligently pick a good source model using the method from Maho et al. (2023), for example.

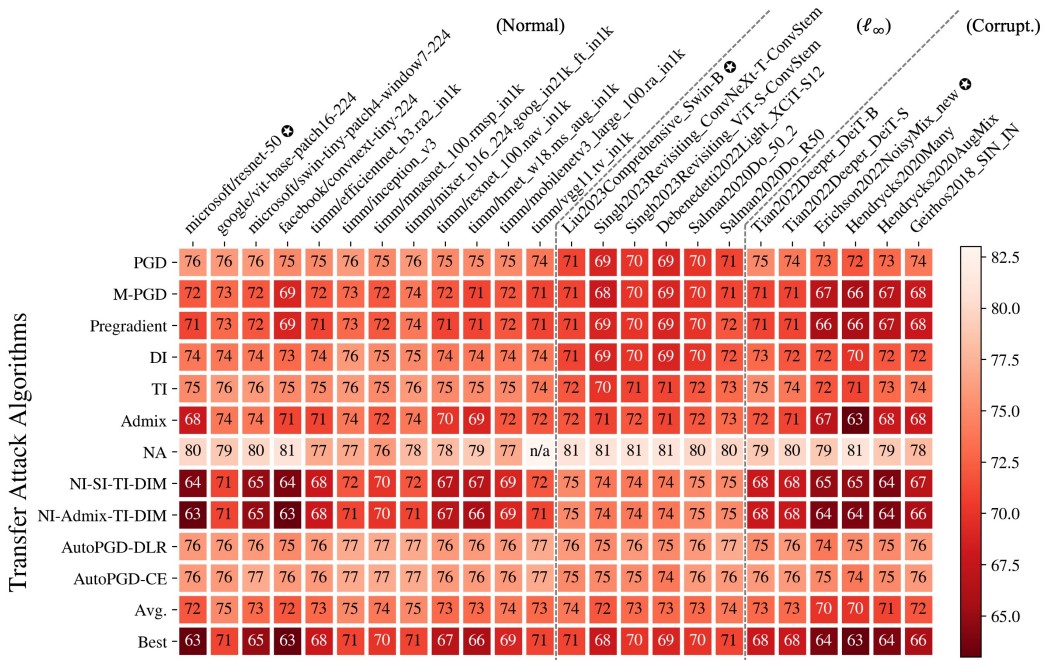

Figure 3: Adversarial accuracy of PUBDEF against 264 transfer attacks (24 source models × 11 attack algorithms) on ImageNet. ✪ denotes the source models this defense is trained against. We cannot produce NA attack on `timm`'s VGG model (shown as "n/a") because of its in-place operation.

| Src. | Algo. | CIFAR-10 | CIFAR-100 | ImageNet |
|------|-------|----------|-----------|----------|
| Seen | Seen | 90.3 | 52.6 | 70.6 |
| Unseen | Seen | 90.3 (−0.0) | 52.6 (−0.0) | 68.6 (−2.0) |
| Seen | Unseen | 90.3 (−0.0) | 50.8 (−1.8) | 63.0 (−7.6) |
| Unseen | Unseen | 88.6 (−1.7) | 50.8 (−1.8) | 63.0 (−7.6) |

Table 2: Adversarial accuracy of PUBDEF under seen/unseen transfer attacks. Seen attacks (seen src. and seen algo.) are the 3–4 attacks that were used to train our defense, unseen attacks are all others from the set of 264 possible attacks. They are categorized by whether the source models (src.) and the attack algorithms (algo.) are seen. All non-PGD attacks are unseen attack algorithms.

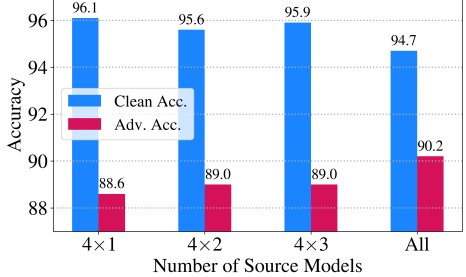

Figure 4: Clean and adversarial accuracy on four PUBDEF models trained with 4 ($4 \times 1$), 8 ($4 \times 2$), 12 ($4 \times 3$), and 24 (All) source models. "$4 \times m$" means $m$ source models are chosen from each of the four groups.

minimal drop in the clean accuracy is one of the most attractive properties of PUBDEF making it far more practical than white-box adversarial training.

**PUBDEF generalizes well to unseen source models and attack algorithms.** Our defense is trained against only the PGD attack and either four (CIFAR-10) or three (CIFAR-100, ImageNet) source models. This amounts to four potential transfer attacks out of 264. Table 2 shows that PUBDEF generalizes incredibly well to the 260 unseen attacks losing only 1.7%, 1.8%, and 7.6% in robustness across the three datasets. This result implies that these 264 transfer attacks may be much more "similar" than the community expected; see Section 7.3.

**PUBDEF should be trained with one source model from each of the four groups.** Table 9 shows an ablation study, where we omit one of the four source models during training. We see that including at least one source model from each of the four groups is important for strong robustness. Including at least one $\ell_2$-robust and at least one corruption-robust model in the source set seems particularly important. Without them, adversarial accuracy drops by 28.5% or 31.7%, respectively. We provide further evidence to support this finding in with a more sophisticated ablation study (Appendix A.5) that controls for the number of source models (Fig. 8).

**Training PUBDEF against more source models is not necessarily better.** Fig. 4 shows that adding more source models (8, 12, or 24) to PUBDEF increases the adversarial accuracy by only ∼1%, and it also decreases clean accuracy by ∼1%. This suggests that our simple heuristic of selecting one source model per group is not only necessary but also sufficient for training PUBDEF.

**A simple loss function suffices.** For CIFAR-10 and CIFAR-100, the RANDOM training loss achieves the best results (Table 10). For ImageNet, ALL is slightly better and is only slightly worse than the best result (DYNAMICLOSS). We use these training methods in all of our evaluations.

## 7 DISCUSSION

### 7.1 ABLATION STUDIES

**Random source model selection.** We experiment with two random methods for choosing the source models for training PUBDEF. In three out of four cases, PUBDEF with the random selection method still outperforms the white-box adversarial training by roughly 10 p.p., but in all cases, it is worse than our default selection method. This result lets us conclude that (i) our simple selection scheme in Section 5.2 is much more effective than random and (ii) all of the source model groups should be represented which is in line with Section 6.2. We refer to Appendix A.5.1 for more details.

**Attacks from ensembles of the public source models.** A more sophisticated adversary could use an ensemble of public models to generate a transfer attack, which has been shown to improve the attack transferability (Liu et al., 2017; Gubri et al., 2022). We experiment with this approach by creating three ensembles[4] of four source models (one model from each group) and generate adversarial samples with all 11 attack algorithms. This results in 33 attack candidates, and then, we report accuracy under the *best* attack among these 33. We found that PUBDEF is robust against such an ensemble attack, and this ensemble attack is no better than an attack constructed from the best single-source model (91.7% vs 88.6% adversarial accuracy). We leave more sophisticated ensemble-based attacks (e.g., Chen et al. (2023b;a)) as future work.

### 7.2 ROBUSTNESS TO WHITE-BOX AND QUERY-BASED ATTACKS

As we design PUBDEF to specifically defend against transfer attacks, we rely on system-level defenses for other types of attacks. Here, we measure the risk against such attacks. First, we find that PUBDEF is not robust against white-box attacks; PGD attack with sufficient steps reduces its accuracy to 0%. This is expected as white-box AT has been the only reliable defense against white-box attacks. Against both soft and hard-label query-based attacks, PUBDEF is also less robust compared to the best white-box AT (Tables 12 and 13). However, once we apply an existing defense against query-based attacks (Qin et al., 2021), PUB-DEF becomes the most robust (Table 3). Specifically, we chose the standard deviation of the additive noise $\sigma \in \{0.01, 0.02\}$ and followed the same evaluation procedure as Qin et al. (2021). We evaluated the defense against the soft-label Square attack (Andriushchenko

| Models | None | Sq-100 | Sq-1k |
|---|---|---|---|
| No Defense ($\sigma$=0) | **96.3** | 36.6 | 0.2 |
| No Defense ($\sigma$=0.01) | 92.0 | 71.0 | 53.2 |
| No Defense ($\sigma$=0.02) | 89.2 | 75.4 | 66.2 |
| White-Box AT ($\sigma$=0) | 85.3 | 77.9 | 67.3 |
| White-Box AT ($\sigma$=0.01) | 85.2 | 80.7 | 76.5 |
| White-Box AT ($\sigma$=0.02) | 85.0 | 81.7 | 78.9 |
| PUBDEF ($\sigma$=0) | 96.1 | 55.2 | 8.8 |
| PUBDEF ($\sigma$=0.01) | 92.6 | 81.4 | 75.6 |
| PUBDEF ($\sigma$=0.02) | 88.9 | **82.1** | **79.8** |

Table 3: Accuracy under Square attack with 100 and 1000 queries. Each model is combined with Qin et al. (2021) with $\sigma$ of 0 (no noise), 0.01, and 0.02.

et al., 2020) with 100 and 1000 queries. The PUBDEF models have both higher clean accuracy (4–7 pp.) and adversarial accuracy (∼1 pp.) than the white-box AT. Adding PUBDEF to this query-based defense boosts the robustness up to 24 pp. (from 66 to 80) against the 1000-query Square attack.

### 7.3 GENERALIZATION AND ADVERSARIAL SUBSPACE

**Surprising generalization.** In Section 6.2, PUBDEF demonstrates excellent robustness against a broad range of TAPM attacks, even the transfer attacks from a source model and/or an attack algorithm that were not trained against. We suspect that the surprising generalization of PUBDEF indicates a low-dimensional structure underlying transfer adversarial examples. We visualize our intuition in Appendix A.6.4. In this section, we investigate this phenomenon more closely.

---

[4]The ensemble averages the logits rather than averaging the losses or the softmax scores; in our past experience, we have found that all three choices yield similar performance.

**Generalization with one source model.** We train four PUBDEF models each against only one source model (not one per group). The adversarial accuracy by groups in Fig. 6 shows an unexpected result: **training against either an $\ell_2$ or corruption source model alone provides above 55% accuracy against the best attack**. Furthermore, training against the $\ell_2$ (resp. corruption) source model provides over 80% robustness against the $\ell_\infty$ (resp. normal) group. This generalization effect does not necessarily hold in reverse (e.g., training against a $\ell_\infty$ source model yields little robustness to the $\ell_2$ group). Some source models are better than others to train PUBDEF with.

To verify the manifold hypothesis, we attempt to quantify it using two metrics: cosine similarity and principal component analysis (PCA). Fig. 5 shows the pairwise cosine similarity values among all 264 attacks aggregated by the four source model groups and averaged over all CIFAR-10 test samples. The cosine similarity is notably higher when comparing adversarial examples generated within the same group in contrast to those across two groups, especially for the $\ell_\infty$ and the $\ell_2$ adversarial training groups (0.23 and 0.24 respectively). The cosine similarity is albeit lower in the normal and the corruption groups. PCA analysis also supports this observation, showing evidence for a low-dimensional linear subspace for the $\ell_\infty$ and the $\ell_2$ groups. We defer the detailed discussion to Appendix A.6.4.

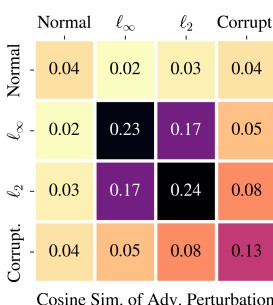

Figure 5: Cosine sim. among pairs of adversarial perturbations by source model group.

### 7.4 PRACTICAL CONSIDERATIONS

PUBDEF is intended to stop a specific class of attacks that are particularly easy to mount, and that are not stopped by any reasonable systems-level defense. However, it has many limitations:

1. PUBDEF is not robust to white-box attacks and is only suitable if model weights can be kept secret. If the model is deployed to users, then attackers can reverse-engineer it to find the weights and mount a white-box attack (Liang et al., 2016; Tencent Keen Security Lab, 2019).
2. A sufficiently dedicated and well-financed attacker can likely train their own surrogate model, e.g., by somehow assembling a suitable training set and paying annotators or using the target model to label it, then mount a transfer attack from this private surrogate, potentially bypassing our defense.
3. We only defend against $\ell_\infty$-norm-bounded attacks. Many researchers have argued persuasively that we also need to consider broader attack strategies (Gilmer et al., 2018; Kaufmann et al., 2023), which we have not examined in this work.

Despite these limitations, PUBDEF has several practical benefits: (1) Minimal drop in clean accuracy: The robustness gain of PUBDEF is (almost) free! This makes it almost ready to deploy in the real world. (2) Low training cost: Adversarial training takes $\sim 2\times$ longer due to the adversarial example generation inside of the training loop. In contrast, PUBDEF training is much faster, as transfer attacks can be pre-generated prior to the training, only need to be done once, and can be done in parallel.

More precisely, the training time of PUBDEF is approximately $16T + 0.2NT$ where $T$ is the time for one epoch of white-box adversarial training and $N$ is the number of epochs of training. In comparison, adversarial training takes $NT$ time. For $N = 50$ epochs, PUBDEF's training time is $26T$ vs $50T$ for adversarial training. The first term in the formula ($16T$) is a one-time cost for generating all the transfer attacks used in training: 4 source models $\times$ 4 instances per training sample $\times$ 10 PGD steps $= 16T$. Regardless of how many PubDef models are trained or how many hyperparameters are swept, this step has to be done once. The second term is the cost of the training loop which depends on the exact loss function used (Section 5.1). For the "Random" loss function, the cost is $0.2T$ per epoch.

## 8 CONCLUSION

In this paper, we propose a pragmatic method for achieving as much robustness as possible, in situations where any more than minimal decrease in clean accuracy is unacceptable. We identify transfer attacks from public source models (TAPM) as a particularly important class of adversarial attacks, and we devise a new method, PUBDEF, for defending against it. Putting everything together yields a plausible defense against adversarial examples by aligning ML defenses with the most feasible attacks in practice that existing systems-level defenses cannot prevent. We hope other researchers will build on these ideas to achieve even stronger protections against adversarial examples.

ACKNOWLEDGEMENTS

This work was supported in part by funds provided by the National Science Foundation (under grant 2229876), the KACST-UCB Center for Secure Computing, the Department of Homeland Security, IBM, the Noyce Foundation, Google, Open Philanthropy, and the Center for AI Safety Compute Cluster. Any opinions, findings, and conclusions or recommendations expressed in this material are those of the author(s) and do not necessarily reflect the views of the sponsors.

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

| Attack Algorithms | Momentum | Augmentation | Feature-Level | Non-CE Loss |
|---|---|---|---|---|
| PGD (Madry et al., 2018) | ✗ | ✗ | ✗ | ✗ |
| M-PGD (Dong et al., 2018) | ✓ | ✗ | ✗ | ✗ |
| Pregradient (Wang et al., 2021b) | ✓ | ✗ | ✗ | ✗ |
| Diverse input (DI) (Xie et al., 2019) | ✗ | ✓ | ✗ | ✗ |
| Translation-invariant (TI) (Dong et al., 2019) | ✗ | ✓ | ✗ | ✗ |
| Admix (Wang et al., 2021a) | ✗ | ✓ | ✗ | ✗ |
| NA (Zhang et al., 2022) | ✗ | ✗ | ✓ | ✗ |
| NI-SI-TI-DIM (Zhao et al., 2022) | ✓ | ✓ | ✗ | ✗ |
| NI-Admix-TI-DIM (Zhao et al., 2022) | ✓ | ✓ | ✗ | ✗ |
| AutoPGD-DLR (Croce & Hein, 2020) | ✗ | ✗ | ✗ | ✓ |
| AutoPGD-CE (Croce & Hein, 2020) | ✗ | ✗ | ✗ | ✗ |

Table 4: All transfer attack algorithms we experiment with in this paper along with the types of techniques they use to improve attack transferability.

# A    APPENDIX

## A.1    DETAILED EXPERIMENT SETUP

### A.1.1    FULL LIST OF PUBLIC SOURCE MODELS

Table 5, Table 6, and Table 7 compile all the source models we use in our experiments for CIFAR-10, CIFAR-100, and ImageNet, respectively. For the normal group, model names that begin with `timm/`, `chenyaofo/`, and `huyvnphan/` are from the `timm` library (Wightman, 2019), Chen (2023), and Phan (2023), respectively. Those that ends with `_local` are models that we train ourselves locally. These include ConvMixer models and several models on CIFAR-100 where the public models are more difficult to find. The remaining models are from Hugging Face's model repository (Face, 2023).

There is no $\ell_2$ group for CIFAR-100 and ImageNet because ROBUSTBENCH does not host any $\ell_2$-adversarially trained models apart from the ones trained on CIFAR-10. As mentioned in Section 6.1, the models are all chosen to have as much diversity as possible in order to ensure that the defenses are evaluated on a wide range of different transfer attacks.

### A.1.2    FULL LIST OF TRANSFER ATTACK ALGORITHMS

All of the 11 transfer attack algorithms we experiment with in this paper are listed in Table 4. Two other attacks that we do not experiment with individually but as a part of the other two combined attacks (NI-SI-TI-DIM, NI-Admix-TI-DIM) are the scale-invariant (SI) (Lin et al., 2020) and the Nesterov (NI) (Lin et al., 2020) attacks. We either take or adapt the attack implementation from Zhao et al. (2022) (https://github.com/ZhengyuZhao/TransferAttackEval). Then, we check our implementation against the official implementation of the respective attacks.

**Notes on some attack implementation.** NA attack was originally designed to use the intermediate features of ResNet-like models so it lacks implementation or instruction for choosing the layer for other architectures. Therefore, we decided to pick an early layer in the network (about one-fourth of all the layers). For the same reason, we also omit NA attack on the source model `Diffenderfer2021Winning_LRR_CARD_Deck` as it is an ensemble of multiple architectures.

### A.1.3    PUBDEF TRAINING

All CIFAR-10/100 models are trained for 200 epochs with a learning rate of 0.1, weight decay of $5 \times 10^{-4}$, and a batch size of 2048. ImageNet models are trained for 50 epochs with a learning rate of 0.1, weight decay of $1 \times 10^{-4}$, and a batch size of 512. We have also swept other choices of hyperparameters (learning rate $\in \{0.2, 0.1, 0.05\}$ and weight decay $\in \{1 \times 10^{-4}, 5 \times 10^{-4}\}$) but did not find them to affect the model's performance significantly. Overall PUBDEF is not more sensitive to hyperparameter choices anymore than a standard training and is less sensitive than white-box adversarial training. All of the models are trained on Nvidia A100 GPUs.

| Groups | Model Names |
|--------|-------------|
| Normal | `aaraki/vit-base-patch16-224-in21k-finetuned-cifar10`
`jadohu/BEiT-finetuned`
`ahsanjavid/conVneXt-tiny-finetuned-cifar10`
`chenyaofo/resnet20`
`chenyaofo/vgg11-bn`
`chenyaofo/mobilenetv2-x0-5`
`chenyaofo/shufflenetv2-x0-5`
`chenyaofo/repvgg-a0`
`huyvnphan/densenet121`
`huyvnphan/inception-v3`
`convmixer_local`
`clip` |
| $\ell_\infty$ | `Wang2023Better_WRN-70-16`
`Xu2023Exploring_WRN-28-10`
`Debenedetti2022Light_XCiT-L12`
`Sehwag2020Hydra` |
| $\ell_2$ | `Wang2023Better_WRN-70-16`
`Rebuffi2021Fixing_70_16_cutmix_extra`
`Augustin2020Adversarial_34_10_extra`
`Rony2019Decoupling` |
| Corruption | `Diffenderfer2021Winning_LRR_CARD_Deck`
`Kireev2021Effectiveness_RLATAugMix`
`Hendrycks2020AugMix_ResNeXt`
`Modas2021PRIMEResNet18` |

Table 5: All public source models for CIFAR-10.

**Data augmentation.** Unlike most training procedures, there are two separate places where data augmentation can be applied when training PUBDEF: the first is when generating the transfer attacks and the second is afterward when training the model with those attacks. For CIFAR-10/100, we use the standard data augmentation (pad and then crop) in both places. For ImageNet, we also use the standard augmentation including random resized crop, horizontal flip, and color-jitter in both of the steps. However, in the second step, we make sure that the resizing does not make the images too small as they would have already undergone the resizing once before. For all of the datasets, we also add CutMix (Yun et al., 2019) to the second step. No extra data or generated data (e.g., from generative models) are used.

**Number of transfer attack instances.** For each pair of $(S, A)$, we generate multiple attacked versions of each training image (by random initialization) to prevent the model from overfitting to a specific perturbation. Unless stated otherwise, we use four versions in total, one of which is randomly sampled at each training iteration. That said, using fewer versions does not significantly affect the performance of our defense. With one version, the adversarial accuracy drops by 0.5% on CIFAR-10 and 0.2% on ImageNet. Using eight versions increases it by 1.2% on CIFAR-10. The clean accuracy seems unaffected by the number of versions. This suggests that when computation or storage is a bottleneck, we can use only one attack version with minimal loss of robustness.

## A.2 ADDITIONAL DESCRIPTION OF THE BASELINES

**EAT** (Tramèr et al., 2018) is an ensemble training technique motivated by the limitations of white-box FGSM training due to their propensity for falling into degenerate global minima that consist of weak perturbations, resulting in weak transfer robustness. EAT is claimed to specifically increase robustness against black-box-based attacks, where members of an ensemble are pre-trained models that serve to augment the data by generating adversarial examples to attack a singular member model.

**DVERGE** (Yang et al., 2020) is a robust ensemble training technique that leverages cross-adversarial training. In this methodology, ensemble members are subject to training on adversarial examples generated from other models present within the same ensemble. The distinction between DVERGE

| Groups | Model Names |
|---|---|
| Normal | `Ahmed9275/Vit-Cifar100`
`MazenAmria/swin-tiny-finetuned-cifar100`
`chenyaofo/resnet20`
`chenyaofo/vgg11-bn`
`chenyaofo/mobilenetv2-x0-5`
`chenyaofo/shufflenetv2-x0-5`
`chenyaofo/repvgg-a0`
`densenet121_local`
`senet18_local`
`inception-v3_local`
`convmixer_local`
`clip` |
| $\ell_\infty$ | `Wang2023Better_WRN-70-16`
`Cui2023Decoupled_WRN-28-10`
`Bai2023Improving_edm`
`Debenedetti2022Light_XCiT-L12`
`Jia2022LAS-AT_34_20`
`Rade2021Helper_R18_ddpm` |
| Corruption | `Diffenderfer2021Winning_LRR_CARD_Deck`
`Modas2021PRIMEResNet18`
`Hendrycks2020AugMix_ResNeXt`
`Addepalli2022Efficient_WRN_34_10`
`Gowal2020Uncovering_extra_Linf`
`Diffenderfer2021Winning_Binary` |

Table 6: All public source models for CIFAR-100.

| Groups | Model Names |
|---|---|
| Normal | `microsoft/resnet-50`
`google/vit-base-patch16-224`
`microsoft/swin-tiny-patch4-window7-224`
`facebook/convnext-tiny-224`
`timm/efficientnet_b3.ra2_in1k`
`timm/inception_v3`
`timm/mnasnet_100.rmsp_in1k`
`timm/mixer_b16_224.goog_in21k_ft_in1k`
`timm/rexnet_100.nav_in1k`
`timm/hrnet_w18.ms_aug_in1k`
`timm/mobilenetv3_large_100.ra_in1k`
`timm/vgg11.tv_in1k` |
| $\ell_\infty$ | `Liu2023Comprehensive_Swin-B`
`Singh2023Revisiting_ConvNeXt-T-ConvStem`
`Singh2023Revisiting_ViT-S-ConvStem`
`Debenedetti2022Light_XCiT-S12`
`Salman2020Do_50_2`
`Salman2020Do_R50` |
| Corruption | `Tian2022Deeper_DeiT-B`
`Tian2022Deeper_DeiT-S`
`Erichson2022NoisyMix_new`
`Hendrycks2020Many`
`Hendrycks2020AugMix`
`Geirhos2018_SIN_IN` |

Table 7: All public source models for ImageNet.

and EAT or adversarial training is the "distillation" loss that encourages a large difference between the intermediate features between models in the ensemble.

**TRS** (Yang et al., 2021) claims to be an improvement over DVERGE. It removes the extra loss term from DVERGE and incorporates $\ell_2$-based regularization into the loss objective function to enhance model smoothness. This regularization penalizes both the cosine similarity between the gradients of all pairs of models in the ensemble and the Euclidean norm of the gradients. Both TRS and DVERGE are evaluated against transfer attacks where they show a superior robustness compared to EAT as well as single-model adversarial training.

## A.3   Simple Game's Equilibrium

The simple game (Section 4.1) can be formulated as the following von Neumann's minimax theorem with a bilinear objective function. We refer the interested readers to Roughgarden (2016) for the full derivation.

**Theorem 1** (von Neumann's minimax theorem with a bilinear function (v. Neumann, 1928)). *Given a "simple game" described above and its payoff matrix $\boldsymbol{R}$, there exists a mixed strategy $\pi_A^*$ for the attacker and a mixed strategy $\pi_D^*$ for the defender such that*

$$r_D^* = \max_{\pi_D \in \Delta^{sa-1}} \min_{\pi_A \in \Delta^{sa-1}} \pi_D^\top \boldsymbol{R} \pi_A = \min_{\pi_A \in \Delta^{sa-1}} \max_{\pi_D \in \Delta^{sa-1}} \pi_D^\top \boldsymbol{R} \pi_A \tag{5}$$

*where $\Delta^{sa-1}$ is the $(sa-1)$-dimensional probability simplex.*

This problem can be solved like any other linear program with polynomial-time algorithms (van den Brand, 2020).

## A.4   PubDef's Loss Function

Below are all the loss functions we have experimented with when training PubDef models. Their comparison is reported in Table 10.

**1. Fixed**: Here, all $\pi_i$'s are fixed to $1/sa$. We experiment with two very similar loss functions: (1) All: The model is trained on all transfer attacks (pairs of $(\mathsf{S}, \mathsf{A})$) simultaneously. This is exactly the same as Eq. (4). (2) Random: Randomly sample one pair of $(\mathsf{S}, \mathsf{A})_i$ at each training iteration.

**2. Top-$k$**: This scheme modifies All by taking, at each iteration of training, the top $k$ pairs of $(\mathsf{S}, \mathsf{A})_i$ that maximize the loss on the current defense model being trained. Effectively, in each batch, we attack the current model weights with all $s \cdot a$ attacks, choose the $k$ most effective attacks, and set their $\pi_i$'s to $1/k$ and the other $\pi_i$'s to 0. For $k = 1$, this is a minimax problem similar to adversarial training, but the maximization is over the choice of transfer attacks instead of the perturbation:

$$\arg\min_{\theta} \ \mathbb{E}_{x,y} \left[ \mathsf{L}(f_\theta(x), y) + \max_{i \in [s \cdot a]} \ \mathsf{L}\left(f_\theta\left(x_{\mathrm{adv},i}\right), y\right) \right] \tag{6}$$

**3. Dynamic weights**: This method can be considered a smooth version of the top-$k$. Instead of setting each $\pi_i$ to either 1 or 0, we dynamically adjust it in proportion to the overall loss or error rate of the attack $(\mathsf{S}, \mathsf{A})_i$. We call these methods DynamicLoss and DynamicAcc respectively. We use an exponential moving average $\mu$ with a decaying factor $\alpha$ to estimate the loss and the accuracy:

$$\mu_i^{t+1} = (1 - \alpha)\mu_i^t + \alpha \mathsf{L}\left(f_\theta\left(x_{\mathrm{adv},i}\right), y\right) \quad \text{(DynamicLoss)} \tag{7}$$

$$\mu_i^{t+1} = (1 - \alpha)\mu_i^t + \alpha \mathbb{E}_{x,y}\left[\mathbb{1}\left\{f_\theta\left(x_{\mathrm{adv},i}\right) = y\right\}\right] \quad \text{(DynamicAcc)} \tag{8}$$

$$\pi_i = \frac{\mu_i}{\sum_{j=1}^{s \cdot a} \mu_j} \quad \text{(Normalize to } [0, 1]) \tag{9}$$

We can normalize the $\pi_i$'s by their sum because both the loss and the error rate are non-negative.

## A.5   Defender's Source Model Selection

In this section, we specifically dive deeper into the source model selection heuristic for training PubDef. We will investigate various effects on the adversarial accuracy under the TAPM transfer attacks when including/excluding certain models.

**Single source model.** We begin by first choosing only one source model from each group. The results reveal that training PubDef with one source model is insufficient for building a good defense. In

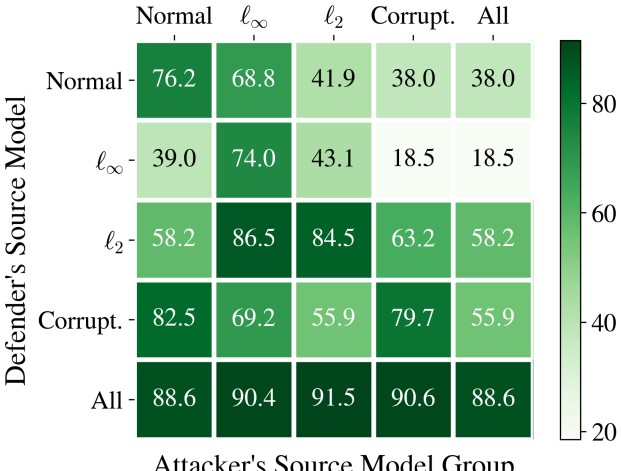

Figure 6: Adversarial accuracy of PUBDEF when trained against only one source model on CIFAR-10. We select one source model from each group, and the adversarial accuracy is also categorized by the attacker's source model groups.

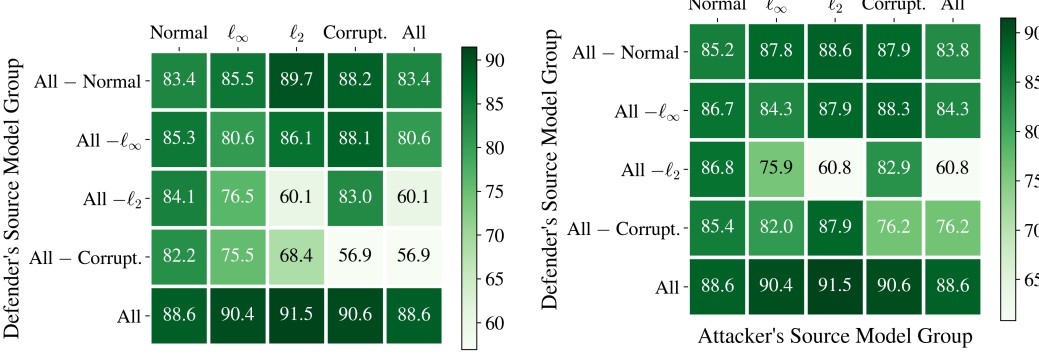

Figure 7: (CIFAR-10) Adversarial accuracy grouped by the attacker's source models (columns). Each row corresponds to a PUBDEF model with one source model from each group *removed* except for the last row where all are included.

Figure 8: (CIFAR-10) Adversarial accuracy grouped by the attacker's source models (columns). Each row corresponds to a PUBDEF model with one source model from each group removed, but unlike Fig. 7, another source model from a different group is added to keep the total number of source models constant (four).

other words, regardless of which source model is chosen, it performs far worse than the default option to include four source models ("All"). That said, the model trained against a singular source model from the $\ell_2$ or the corruption group exhibits surprisingly high resilience even to the best transfer attack, achieving the adversarial accuracy of over 55%.

**Excluding one source model group.** Fig. 7 is an extension to Table 9 which shows the adversarial accuracy from the best attack from each source model group in addition to the overall best. Here, by looking at the diagonal entries, we can see that removing a source model from one group during PUBDEF training reduces the robustness against the transfer attack from the same group. Note that the overall best transfer attack always comes from the missing group (i.e., the diagonal entries are equal to the last column). The degradation is minor for the normal and the $\ell_\infty$ groups but very large for the $\ell_2$ and the corruption groups. This implies that for certain reasons, adversarial examples generated on source models from the normal and the $\ell_\infty$ groups are "subsets" of the ones from the other two groups because this generalization phenomenon happens only in one direction but not both.

| Defenses | CIFAR-10 | | CIFAR-100 | | ImageNet | |
|---|---|---|---|---|---|---|
| | Clean | Adv. | Clean | Adv. | Clean | Adv. |
| Best white-box adv. training | 85.3 | 68.8 | 68.8 | 32.8 | 63.0 | 36.2 |
| PUBDEF (all random) | 95.2 (+9.9) | 62.9 (−5.9) | 75.4 (+6.6) | 44.0 (+11.2) | - | - |
| PUBDEF (random per group) | 95.0 (+9.7) | 77.3 (+8.5) | 75.3 (+6.5) | 44.2 (+11.4) | - | - |
| **PUBDEF (default)** | 96.1 (+10.8) | 88.6 (+19.8) | 76.2 (+7.4) | 50.8 (+18.0) | 78.5 (+15.5) | 62.3 (+26.1) |

Table 8: Clean and adversarial accuracy of PUBDEF against the best attack under the TAPM threat model. As an ablation study, we compare the baseline as well as our default PUBDEF to PUBDEF when the defender's source models are randomly selected.

We conduct another experiment that controls for the fact that the number of source models used to train PUBDEF reduces from four (the default setting where all source models are used) to three in the earlier experiments. The results are shown in Fig. 8. Here, in addition to removing one source model, we add another source model from a different group to the pool. For example, in the first row, the normal model is removed, and then we add a model from each of the remaining groups to create three different PUBDEF models ($\{\ell_\infty, \ell_\infty, \ell_2, \text{corrupt.}\}, \{\ell_\infty, \ell_2, \ell_2, \text{corrupt.}\}, \{\ell_\infty, \ell_2, \text{corrupt.}, \text{corrupt.}\}$). We then compute the adversarial accuracy of all three new models and report the best one. The conclusion above remains unchanged, but the gap in the adversarial accuracy becomes particularly smaller for the corruption group.

### A.5.1 RANDOM SOURCE MODEL SELECTION

In this section, we provide detailed experiments and results on the random source model selection method for training PUBDEF. We train two new sets of 30 PUBDEF models each where the source models are randomly sampled in two different ways: (A) by model and (B) by group. When randomly sampling *by model*, all source models have an equal probability of being selected. This means that more normal models may be selected as they are over-represented (12 vs. 4/6 for the other groups). When sampling *by group*, we uniformly sample one out of four groups and then uniformly sample one source model from the group. This accounts for the difference in group sizes. The total number of source models is still fixed at four for CIFAR-10 and three for CIFAR-100.

Table 8 compares the two random model selection methods to the default one. Here, "all random" corresponds to the sampling by group, and "random per group" is the by-group sampling but without replacement so all groups are represented by one model. Choosing a random source model from each group (i.e., set B) results in a drop in the adversarial accuracy by 11% on CIFAR-10 and 7% on CIFAR-100 on average (Appendix A.5). However, they are still 9% and 11% higher than the baseline. On the other hand, set A performs worse than the baseline by 6% on CIFAR-10 and similarly to set B on CIFAR-100. The discrepancy between the two datasets on set A can be explained by the fact that the probability of choosing exactly one model from each group is only 7% on CIFAR-10 but 21% on CIFAR-100.

Next, we use these two sampling methods to perform additional ablation studies. By randomly sampling the source models, we can make a general statement about the effect of each source model group in PUBDEF independent of the exact choice of the source models.

**Importance of each source model group.** We train 64 PUBDEF models with one, two, three, and four source model groups (16 each). The source models are sampled by groups without replacement. Fig. 9a and Fig. 9b show the adversarial accuracy by the source model group similarly to Fig. 7 but with the random selection described above. Fig. 9a categorizes PUBDEF by which source model group is *excluded* whereas Fig. 9b categorizes by which source model group is *included*. The result from Fig. 9a very much agrees with Fig. 7 and Fig. 8. Fig. 9b also corroborates with this observation by confirming that PUBDEF models that include the $\ell_2$ group outperforms the others quite significantly.

**Number of total source models.** Fig. 10 displays both the clean and the adversarial accuracy when PUBDEF is trained with different numbers of source models. This is similar to Fig. 4 in the main paper as well as Fig. 6, but the only difference is that the source models here are randomly selected. Similar to the above result, the random selection allows us to make a more generalized statement about the *source model groups* without being strictly tied to the exact choice of the source models.

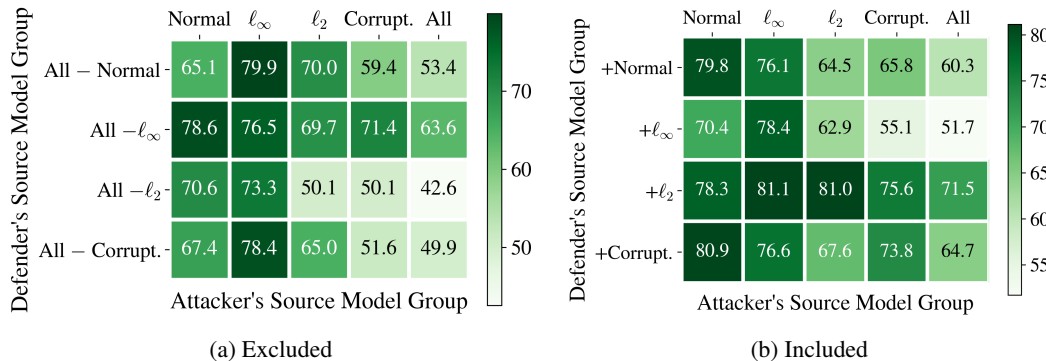

Figure 9: (CIFAR-10) Adversarial accuracy of PUBDEF with the random source model selection (by group). Similar to Fig. 7, we transfer attacks by the source model groups. The left plot categorizes PUBDEF by the excluded group while the right plot categorizes by the included group.

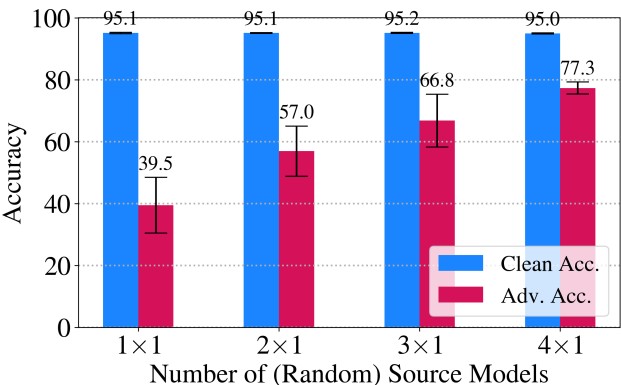

Figure 10: Clean and adversarial accuracy under the best attack on our defense with varying numbers of *randomly chosen* source models for training PUBDEF. $k \times 1$ denotes that we (randomly) pick one model from each of the $k$ categories. The error bars denote $t$-distribution 95%-confidence interval.

This plot demonstrates that the adversarial accuracy against the best transfer attack improves as PUBDEF is trained against more source models, but the return is diminishing. On the other hand, the clean accuracy is completely unaffected and varies in a very narrow range.

## A.6   ADDITIONAL RESULTS

### A.6.1   COMPARISON TO OTHER ENSEMBLE BASELINES

We have described the ensemble defenses including DVERGE and TRS in Section 2 and Appendix A.2. Here, we will introduce another defense we built on top of TRS to make it even more suitable as a defense against transfer attacks. We call this scheme "Frozen TRS." The main idea behind TRS as well as DVERGE is to make the models in the ensemble diverse and different from one another. However, in the TAPM threat model, we want to make our defense "different" from the public models as much as possible. With this in mind, we apply the regularization terms from TRS to a set of models comprised of both the defense we want to train and the public source models we want to train against. But unlike TRS, we are only training one defense model while holding all the public models fixed (hence "frozen").

Table 11 compares all the ensemble-based defenses including the previously described Frozen TRS. It is evident that all the ensembles are outperformed by the single-model white-box adversarial training as well as PUBDEF. Frozen TRS performs better than the normal TRS but worse than the adversarially trained TRS. We believe that TRS' regularization is effective for diversifying the models' gradient directions, but it takes an indirect path to improve the robustness against transfer

| Defender Src. Model Groups | Clean | Adv. |
|---|---|---|
| All groups | **96.1** | **88.6** |
| All groups but normal | 95.4 | 83.4 (−5.2) |
| All groups but $\ell_\infty$ | 95.3 | 80.6 (−8.0) |
| All groups but $\ell_2$ | 95.0 | 60.1 (−28.5) |
| All groups but corruption | 94.9 | 56.9 (−31.7) |

Table 9: Effects on accuracy when excluding one (out of four) defender's source models from PUBDEF trained on CIFAR-10.

| Defense Loss Function | CIFAR-10 | | CIFAR-100 | | ImageNet | |
|---|---|---|---|---|---|---|
| | Clean | Adv. | Clean | Adv. | Clean | Adv. |
| RANDOM | **96.1** | 88.6 | **76.2** | 50.8 | **79.0** | 58.6 |
| TOP-1 | 95.3 | 87.0 | 73.9 | 50.9 | 78.2 | 57.3 |
| TOP-2 | 95.8 | 86.1 | 74.2 | **51.9** | 78.4 | 60.6 |
| ALL | 96.0 | 86.8 | 73.5 | 50.3 | 78.5 | 62.3 |
| DYNAMICACC | 95.2 | **88.9** | 74.0 | 51.5 | 78.6 | 62.8 |
| DYNAMICLOSS | 95.6 | 88.4 | 73.8 | 51.1 | 78.6 | **63.0** |

Table 10: Clean and adversarial accuracy of our defense with different training methods (Section 5).

| Defenses | Clean Acc. | Adv. Acc. |
|---|---|---|
| Best white-box adversarial training | 85.3 | 68.8 |
| TRS | 90.7 | 30.1 |
| TRS + adversarial training | 86.9 | 66.7 |
| DVERGE | 88.6 | 33.4 |
| DVERGE + adversarial training | 87.6 | 59.6 |
| Frozen TRS | 86.1 | 49.9 |
| **PUBDEF** | **96.1** | **88.6** |

Table 11: Comparison of PUBDEF to all the ensemble-based defenses on CIFAR-10.

attacks. Training on the transfer attacks themselves like PUBDEF is more straightforward and so produces a better defense. An interesting future direction is to combine TRS (or other forms of gradient regularization) with PUBDEF.

### A.6.2 ADVERSARIAL ACCURACY ON ALL TRANSFER ATTACKS

In this section, we include the figures containing adversarial accuracy against all the transfer attacks on CIFAR-10 (Fig. 11) and CIFAR-100 (Fig. 12), similarly to Fig. 3 for ImageNet in the main paper. NA attacks are excluded and marked as "n/a" for the ensemble-based model (i.e., `Diffenderfer2021Winning_LRR_CARD_Deck`) because the intermediate feature is more difficult to specify. Some interesting observations:

- PGD and M-PGD are among the best attack algorithms across all of the datasets despite their simplicity compared to the rest of the attacks.
- NA attack is the weakest attack on both CIFAR-100 and ImageNet but the strongest on CIFAR-10.
- The source models from the $\ell_\infty$ group produce stronger transfer attacks than the other groups. The trend is particularly strong on CIFAR-100 but weaker on CIFAR-10 and ImageNet.

### A.6.3 ROBUSTNESS AGAINST QUERY-BASED ATTACKS

We include additional results on the models under query-based attacks. Table 12 shows accuracy under Square attack, a soft-label (or score-based) query-based attack without any system-level defense in place. Table 13 reports accuracy under hard-label (or decision-based) query-based attack also without any system-level defense. Here, the white-box adversarially trained models are more robust than PUBDEF. However, after applying a system-level defense like Qin et al. (2021), PUBDEF has higher accuracy and robustness as shown in Section 7.2.

### A.6.4 DETAILS ON THE ADVERSARIAL SUBSPACE EXPERIMENTS

We start by providing more intuition by visualizing a space of the transfer attack around a training sample $x$. Fig. 13 compares the attack surfaces under the white-box and the TAPM threat model. The white-box attack is inherently more difficult to solve as the defender must train the model to be robust at any point in the $\ell_p$-norm ball of a given radius. On the other hand, under the TAPM

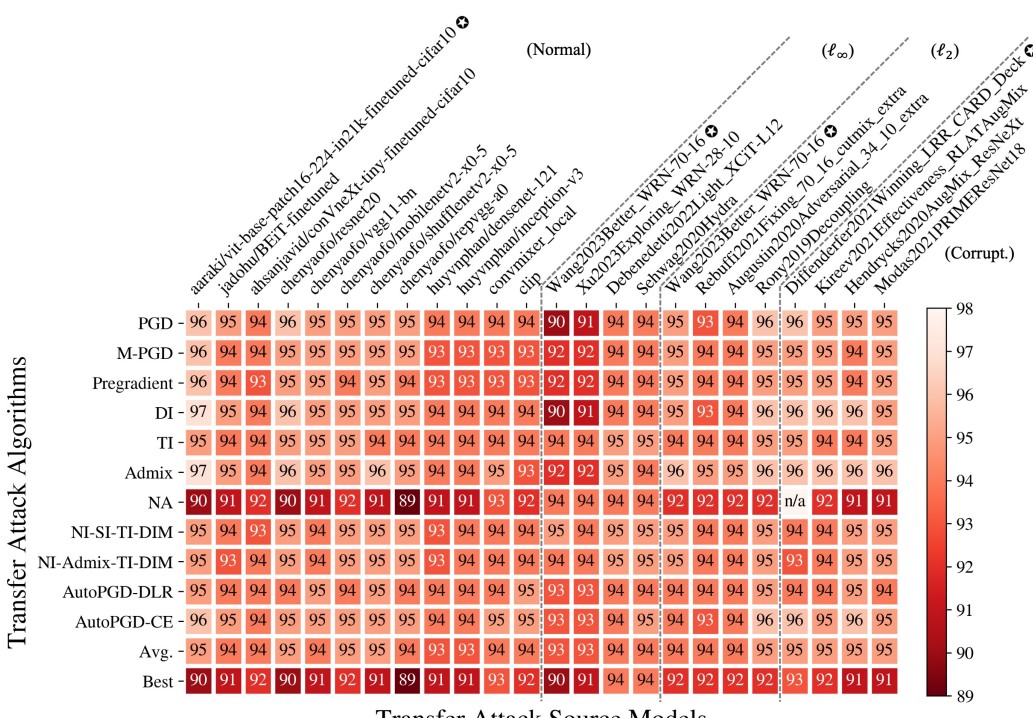

Figure 11: Adversarial accuracy of PUBDEF against 264 transfer attacks (24 source models × 11 attack algorithms) on CIFAR-10. ✪ denotes the source models this defense is trained against.

| Models | CIFAR-10 | | CIFAR-100 | | ImageNet | |
|---|---|---|---|---|---|---|
| | Sq-100 | Sq-1000 | Sq-100 | Sq-1000 | Sq-100 | Sq-1000 |
| No Defense | 36.6 | 0.2 | 9.1 | 1.8 | 49.9 | 17.9 |
| Best White-Box AT | **77.9** | **67.3** | **55.9** | **41.0** | **58.4** | **57.6** |
| PUBDEF | 55.2 | 8.8 | 13.5 | 0.3 | 55.1 | 32.3 |

Table 12: Accuracy of the models under Square Attack (soft-label query-based) with 100 and 1000 queries. Here, no system-level defense is applied.

setting, there are only finitely many attacks, and so they will always "occupy" a smaller volume of the ball. Put differently, they can only span a linear subspace of $\min\{s \cdot a, d\}$ dimensions where $s \cdot a$ is the number of all potential transfer attacks defined in Section 3 and $d$ is the input dimension where usually $d \gg sa$.

Fig. 14 potentially explain why PUBDEF generalizes well to attacks not seen during training. When all the attacks lie in a low-dimensional subspace or form a cluster around a small subsection of the $\ell_p$-norm ball, the sample complexity is also low. Hence, training on a few attacks is sufficient for capturing this adversarial subspace. On the other hand, there can be two potential failure cases of TAPM if the transfer attacks do not actually form a low-dimensional subspace. First, the model can overfit to the training attacks and does not generalize to the unseen. The second case can happen when the model does not necessarily over to the training attack and tries to learn to be robust in the entire $\ell_p$-norm ball. In this case, TAPM could perform similarly to white-box adversarial training, and we may see no benefit as it suffers the same pitfalls as white-box adversarial training. Based on our results, it seems that TAPM may experience the first failure case on a subset of the samples, creating a small generalization gap between the seen and the unseen attacks as shown in Table 2.

**Cosine similarity.** Here, we include additional observations that were not mentioned in Section 7.3. From Fig. 5, there is also a relatively strong similarity (0.17) between the $\ell_\infty$ and the $\ell_2$ groups

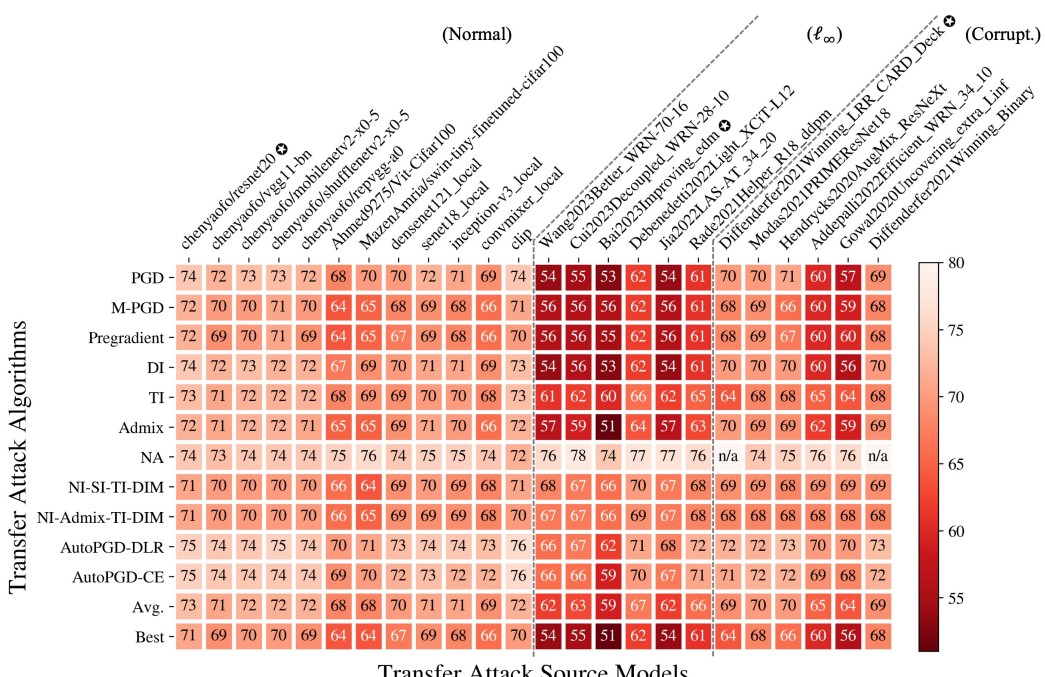

Figure 12: Adversarial accuracy of PUBDEF against 264 transfer attacks (24 source models × 11 attack algorithms) on CIFAR-100. ✪ denotes the source models this defense is trained against.

| Models | CIFAR-10 (HSJ) | CIFAR-100 (HSJ) | ImageNet (HSJ) | ImageNet (GeoDA) |
|---|---|---|---|---|
| No Defense | 0.0276 | 0.0208 | **0.1995** | 0.1186 |
| Best White-Box AT | **0.1542** | **0.1518** | 0.1969 | **0.1662** |
| PUBDEF | 0.0466 | 0.0397 | 0.1814 | 0.1604 |

Table 13: Mean adversarial perturbation norm ($\ell_\infty$) found by two hard-label query-based attacks, HSJ (Chen et al., 2020a) and GeoDA (Rahmati et al., 2020), with 1000 queries. Higher means the model is more robust.

compared to the other cross-group pairs. This corroborates the earlier result as well as an observation made by Croce & Hein (2021) that robustness transfers among different $\ell_p$-norms. However, this result does not explain generalization among normally trained models, as the cosine similarity within the group remains low (0.04). The low similarity might be due to the fact that there are 12 source models in the normally trained group instead of four in the others which implies higher diversity within the normally trained group as well as the transfer attacks generated from them.

**PCA.** In addition to cosine similarity, we fit PCA on the adversarial perturbation in an attempt to verify whether they lie in a low-dimensional linear subspace. This analysis is inspired by the notion of "adversarial subspace" done by Tramèr et al. (2017). First, we randomly choose a test sample as well as all 264 adversarial examples generated from it. We put these adversarial examples into five groups by the source models (including all). Then, we fit PCA to the attacks in each of these groups. If the adversarial examples do lie on a low-dimensional linear manifold, then we should expect that most of the variance is explained by only the first few principal components.

Fig. 15 shows the explained variance plot for three random CIFAR-10 samples, and from this figure, we can see some evidence for the low-dimensional linear subspace for the robust training groups. In the $\ell_\infty$, $\ell_2$, and corruption groups, the first principal component already explains about 20% of all the variance. For $\ell_\infty$ and $\ell_2$, 90% of the variance is explained by approximately 15 dimensions. This result supports our observation based on the cosine similarity: the $\ell_\infty$ and the $\ell_2$ groups seem

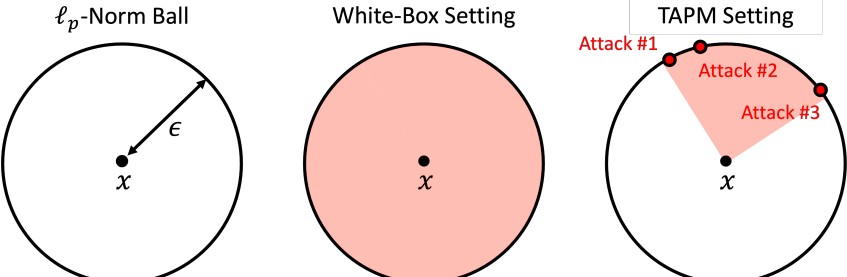

Figure 13: Schematic comparing TAPM to the white-box threat model. **Left**: an $\ell_p$-norm ball with radius $\epsilon$ around a clean input sample $x$. **Middle**: the white-box threat model assumes that attacks can lie anywhere inside the ball and so the defense has to protect the entire ball (red highlight). **Right**: the TAPM threat model expects only a finite set of transfer attacks that may lie on a small-dimensional manifold.

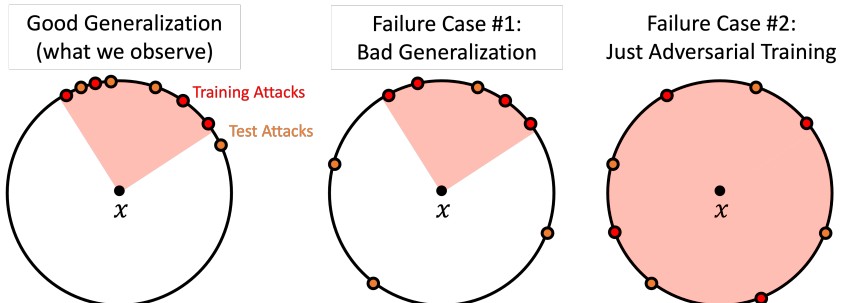

Figure 14: Schematic explaining the generalization phenomenon of PUBDEF. **Left**: the observed case where PUBDEF generalizes well to unseen attack. **Middle**: the first potential failure case where PUBDEF overfits to the attacks used during training. **Right**: the second failure case where PUBDEF does not overfit but does not perform better than adversarial training.

to cluster most tightly (high cosine similarity) followed by the corruption group and lastly by the normal group.

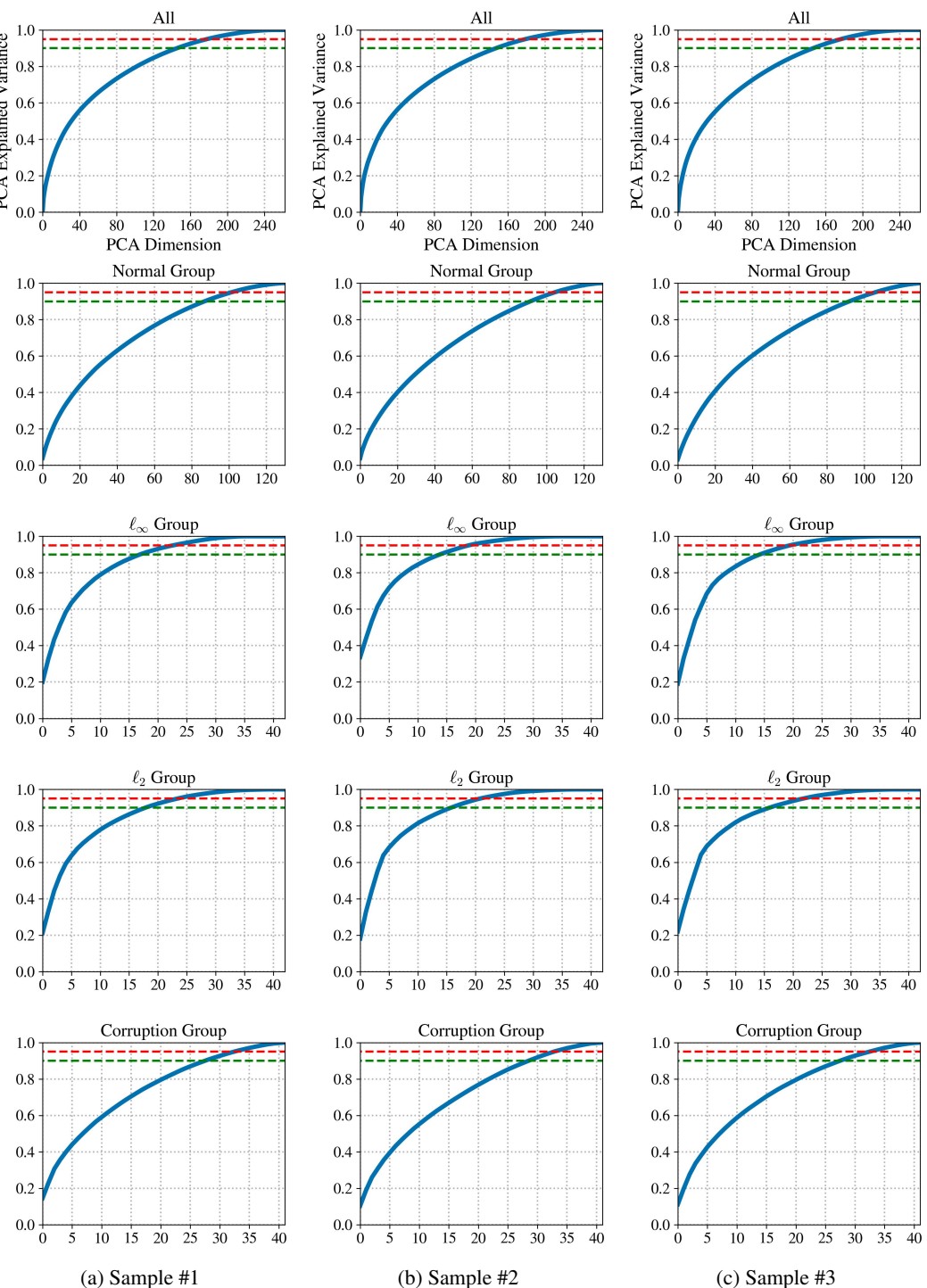

Figure 15: PCA explained variance as a function of dimension (number of the principal components) of the transfer attacks for three random CIFAR-10 samples accumulated by the source model groups. All attack algorithms were used. The green and the red dashed lines denote 90% and 95% of the variance.

