# OpenReview forum: "PubDef: Defending Against Transfer Attacks From Public Models"
_ICLR.cc/2024/Conference — ICLR 2024 poster_

### Official Review · Reviewer_CVr6 · 2023-10-20

**Soundness:** 3 good
**Presentation:** 4 excellent
**Contribution:** 3 good
**Rating:** 8
**Confidence:** 3

**Summary:**

This paper proposes a pragmatic defence against a newly proposed practical threat model - namely adversaries utilising transfer attacks via publicly available surrogate models (with no ability to fine-tune models or perform query-based attacks). The defence is motivated by a game-theoretic perspective, and evaluated against a range of public models, attack algorithms and datasets.

**Strengths:**

A very well written and presented paper. The game-theoretic motivation is clear and provides a reasoned foundation for the approach taken - though that approach does not fully solve the game-theory problem as posed, the framework at least justifies the concept of the approach as more than just arbitrary/ad hoc.

Thorough experimental results including extensive ablation studies. The investigations, for example, of the effects of removing various groups of models on the accuracy was interesting. Further, investigating the cosine similarity of perturbations between attacks was suggestive of further structure to be explored in future work.

The candidate defence outperforms SOTA by a reasonable margin, and provides a practical mechanism in contrast to SOTA adversarial training. I think because of this practical performance requirement for their approach with still SOTA or better results, it could be a useful technique to use in practise or to consider developing further.

**Weaknesses:**

The "slogan" is not very catchy! :-)

**Questions:**

In section 7.3, PUBDEF's vulnerability to white-box and surrogate attacks is mentioned. It would have been good to see some results on this if possible in order to give an idea of the degradation of performance that such weakening of the threat model would provide, i.e. effectively an ablation study on the threat model assumptions.

Is there some value to considering a case in which the set of publicly available models visible to the attacker and defender are different. Maybe there are communities not visible to the defender, or perhaps it could be because of time-based issues (e.g. defender model generated and frozen but subsequent public models appear which are available to the attacker)?

---

> ### Author Response · Authors · 2023-11-17
> **Author response to Reviewer CVr6**
>
> Thank you for taking the time to review our paper. We really appreciate your comments and constructive feedback. Please see the answers and clarifications to your questions below.
>
> ### White-box attacks
>
> > In section 7.3, PUBDEF's vulnerability to white-box and surrogate attacks is mentioned. It would have been good to see some results on this if possible in order to give an idea of the degradation of performance that such weakening of the threat model would provide, i.e. effectively an ablation study on the threat model assumptions.
>
> We appreciate the suggestion. Against white-box attacks, PubDef is no better than an undefended model (~0% accuracy against a 100-step white-box PGD attack). We will add this statement in our revision.
>
> We would like to also take the chance to re-emphasize that a core argument of our paper is that it is unnecessary in practice to make every model robust against white-box attacks. Instead, the defender should identify the most likely threats to their system and focus on those. If the model has to be completely public, then robustness against white-box attacks is necessary. Otherwise, using white-box adversarial training would be overkill, like “using a sledgehammer to crack a nut.”
>
> ### Unseen public models
>
> > Is there some value to considering a case in which the set of publicly available models visible to the attacker and defender are different. Maybe there are communities not visible to the defender, or perhaps it could be because of time-based issues (e.g. defender model generated and frozen but subsequent public models appear which are available to the attacker)?
>
> The paper shows that not much robustness is lost if a defender is unaware of some public models that the attacker is aware of. Please see the discussion of unseen source models in Sections 6.2, 7.2, Table 2, Figure 4. To summarize, the accuracy under attack drops by less than 2 percentage points on CIFAR-10/100 and by 8 points on ImageNet under unseen attacks (unseen attack algorithm and unseen source models). For any other table or figure, we always report the accuracy against the strongest attack including both seen and unseen. In practice, we believe it is likely that defenders can identify most or all publicly available models.
>
> That said, we have conducted an ablation study where **the defender only sees one random public model from each group.** See Table 9 in Appendix A.4.1. Here, PubDef is still significantly better than all the baseline models; on CIFAR-10, it has 10 and 9 percentage points higher clean and adversarial accuracy (7 and 11 pp. for CIFAR-100) than the best white-box adversarially trained models. We note that choosing randomly is completely unrealistic and is the worst possible strategy for the defender. There is no reason to believe that the defender would just randomly choose the source models to train against, so we expect that in practice the defense will perform significantly better than this, even if some models are not known to the defender.
>
> Thank you for reading our rebuttal. Please don’t hesitate to let us know if there are other questions or concerns we have not addressed.

---

> > ### Comment · Reviewer_CVr6 · 2023-11-19
> >
> > Thank you so much for your responses. I retain my original overall review score.

---

> > > ### Author Response · Authors · 2023-11-22
> > >
> > > Thank you for taking the time to read and acknowledge our response!

---

### Official Review · Reviewer_nPyj · 2023-10-22

**Soundness:** 2 fair
**Presentation:** 3 good
**Contribution:** 2 fair
**Rating:** 3
**Confidence:** 5

**Summary:**

This paper proposes a defense against transfer-based evasion attacks. The defense uses publicly-available pretrained models to create adversarial attacks and incorporates these attacks in the training process of the defended model. The process is similar to adversarial training, however it uses adversarial examples from (multiple) other robust models rather than from the trained model itself.

**Strengths:**

+ introduction is well written
+ formulation is clear

**Weaknesses:**

- evaluation is not convincing
- no analysis on how expensive the method is, compared, e.g., to adversarial training

**Questions:**

**Evaluation is not convincing, missing: white-box evaluation, query-based evaluation, other norms, targeted attacks, strong ensemble transfer attacks**

The defense should be tested fully, also against white-box evasion attacks. This is motivated by the fact that the authors are trying to make the model robust, hence they should demonstrate the robustness against worst-case adversaries. Otherwise, there is a high risk that the defense might be broken by stronger attacks. As the approach is similar to AT, the authors should demonstrate that the decision boundary of the model is in fact becoming more robust to unseen adversarial attacks and perturbations, thus they should also test gradient-based attacks against the model itself. Additionally, the authors should test against an attack that is generated ensembling the gradients of multiple models, which should improve transferability and also test effectively the robustness claim that the authors are making.
The model should also be tested with variations of the attacks, e.g., with PGD with logit loss, or against APGD with DLR loss. The authors should demonstrate that the defense generalizes against other unseen attacks and variations, to really claim robustness. The defense should be robust regardless of the method used to find the adversarial attacks.
The model should also be tested against query-based attacks to properly validate the fact that it is not suffering from gradient masking problems.
Finally, all the parameters of the attacks should be specified to ensure they are conducted properly and they are not suffering from optimization issues.

**No analysis on how expensive the method is, compared, e.g., to adversarial training**

The authors should discuss how expensive is to train a model with this technique, compared, e.g., with AT. Additionally, they should specify how costly is to add further heuristics.
For example, the authors use a heuristic to choose the models to use for creating the transfer attacks. However, this heuristic seems expensive, as it requires to compute the adversarial accuracy (in transfer) against all available models. Does this require to launch a full evaluation against all models?


**Clarifications needed**

- the authors state that they propose "using all available system-level defenses", however the presented method is clearly a ML-level defense. The authors should clarify this aspect, as it is confusing to read and there is no discussion in the rest of the paper on which system-level defenses are mounted on the model, or if they are mounted at all.

- the authors state that the drawback of AT is that it degrades clean accuracy, however this does not seem to be the case https://robustbench.github.io. The authors should clarify this aspect by detailing this statement.

- The method assumes that the defended is aware of the same set of public models as the attacker. However, with the proliferation of public models (also supported in the introduction of the paper), this might not be realistic to assume that the defender is aware of all possible models available. Furthermore, assuming that the attacker has $s \dot a$ attack strategies might bring an excessive number of combinations. The authors should clarify these statements and propose solutions for these limitations. In fact, when the authors claim that the proposed approach "achieves over 90% accuracy against all four transfer attacks", they should also disclose that this holds only when the set of target models is the same as the source models $T = S$.

---

> ### Author Response · Authors · 2023-11-17
> **Author response to Reviewer nPyj (1/3)**
>
> Thank you for taking the time to review our paper. We really appreciate your comments and constructive feedback. Please see the answers and clarifications to your questions below.
>
> ### Unseen attacks and public models
>
> > The authors should demonstrate that the defense generalizes against other unseen attacks and variations, to really claim robustness.
>
> > The method assumes that the defended is aware of the same set of public models as the attacker.
>
> **The paper shows that not much robustness is lost if a defender is unaware of some public models that the attacker is aware of.** This surprising effectiveness or generalization is a result of our technique to choose one source model from each group. Please see the discussion of unseen source models in Sections 6.2, 7.2, Table 2, Figure 4. To summarize, the accuracy under attack drops by less than 2 percentage points on CIFAR-10/100 and by 8 points on ImageNet under unseen attacks (unseen attack algorithm and unseen source models). For any other table or figure, we always report the accuracy against the strongest attack including both seen and unseen. In practice, we believe it is likely that defenders can identify most or all publicly available models.
>
> That said, we have conducted an ablation study where **the defender only sees one random public model from each group.** See Table 9 in Appendix A.4.1. Here, PubDef is still significantly better than all the baseline models; on CIFAR-10, it has 10 and 9 percentage points higher clean and adversarial accuracy (7 and 11 pp. for CIFAR-100) than the best white-box adversarially trained models. We note that choosing randomly is completely unrealistic and is the worst possible strategy for the defender. There is no reason to believe that the defender would just randomly choose the source models to train against, so we expect that in practice the defense will perform significantly better than this, even if some models are not known to the defender.
>
> ### White-box evasion attacks
>
> > The defense should be tested fully, also against white-box evasion attacks. This is motivated by the fact that the authors are trying to make the model robust, hence they should demonstrate the robustness against worst-case adversaries.
>
> There appears to be a misunderstanding. The paper makes no claims about the robustness of PubDef against white-box attacks nor about the decision boundary. Against white-box attacks, PubDef is no better than that of an undefended model (0% accuracy against white-box attacks).  PubDef is not designed to provide security against white-box attacks and is not appropriate in scenarios where white-box attacks are possible, so such attacks are beyond the scope of this paper. We do not claim "robustness" in general; rather, we claim robustness against TAPM attacks. Achieving robustness against all white-box attacks, without significant loss of clean accuracy, is an open problem that has yet to be achieved despite concerted effort from the community.
>
> We would like to re-emphasize that a core argument of our paper is that **it is not necessary in practice to make every model robust against white-box attacks.** Instead, the defender should identify the most likely threats to their system and focus on those. If the model has to be completely public, then robustness against white-box attacks is necessary. Otherwise, using white-box adversarial training is overkill, like “using a sledgehammer to crack a nut.”
>
> We believe that a more common situation is that the model does not need to be public. In that case, we can use systems-level defenses to make white-box attacks, query-based attacks, and transfer attacks from non-public models unattractive to an attacker, so that the top priority is to provide robustness against TAPM attacks; and we show that we are able to achieve a surprising level of robustness against such attacks, with low impact on clean accuracy. This is, as far as we are aware, a new insight, and it has significant implications for how to defend models in practice.
>
> ### Variations of attacks
>
> > The model should also be tested with variations of the attacks, e.g., with PGD with logit loss, or against APGD with DLR loss.
>
> There is already extensive research literature on variations of transfer attacks, and we experimented with 11 state-of-the-art schemes in the literature (which also includes APGD with DLR loss). We ensure that they are diverse by choosing ones that employ different mechanisms for improving transferability. While it is always possible to try even more variations, it is not our intention to innovate in transfer attacks, nor do we believe it to be necessary in our case. We also found that our defense generalizes well to “unseen” transfer attack algorithms that it wasn't trained against. See Sections 6.2 and A.5.2 (supplementary material).

---

> > ### Author Response · Authors · 2023-11-17
> > **Author response to Reviewer nPyj (2/3)**
> >
> > ### Transfer attacks from an ensemble
> >
> > > Additionally, the authors should test against an attack that is generated ensembling the gradients of multiple models, which should improve transferability and also test effectively the robustness claim that the authors are making.
> >
> > We already evaluated PubDef against transfer attacks generated with gradients from an ensemble of multiple models. We gather three different ensembles and use all the 11 attack algorithms. Please see “Attacks from ensembles of the public source models” in Section 7.1 for more details. To summarize, the best attack from the ensembles turns out to be weaker than the best attack from a single model (92% vs 89% adversarial accuracy on PubDef on CIFAR-10).
> >
> > ### Attack parameters
> >
> > Please see Section A.1.2 (supplementary material). For evaluation, we set the common attack parameters as follows: step size of 0.001, 100 steps, and 3 random restarts. To generate adversarial examples for training, we use 10-step PGD with step size $\epsilon/4$ and 1 restart. We note that, unlike white-box adversarial training, the transfer attack has to be generated only once outside of the training loop (see the cost of training section below). For the attack-specific parameters, we follow the existing implementation (either an official one or one from [this survey paper](https://github.com/ZhengyuZhao/TransferAttackEval)) to ensure that the parameter choices are reasonable and already tested. We will include this additional detail in the appendix.
> >
> > ### Cost of training
> >
> > > The authors should discuss how expensive is to train a model with this technique, compared, e.g., with AT. Additionally, they should specify how costly is to add further heuristics.
> >
> > **PubDef is about 2$\times$ less expensive than white-box adversarial training.** The training time of PubDef is approximately $16T + 0.2NT$ where $T$ is the time for one epoch of white-box adversarial training and $N$ is the number of epochs of training. In comparison, adversarial training takes $NT$ time. For $N=50$ epochs, PubDef's training time is $26T$ vs $50T$ for adversarial training.
> >
> > The first term in the formula ($16T$) is a one-time cost for generating all the transfer attacks used in training: 4 source models * 4 instances per training sample * 10 PGD steps = 16T. Regardless of how many PubDef models are trained or how many hyperparameters are swept, this step has to be done once. The second term is the cost of the training loop which depends on the exact loss function used (Section 5.1). For the “Random” loss function, the cost is 0.2T per epoch.
> >
> > > However, this heuristic seems expensive, as it requires to compute the adversarial accuracy (in transfer) against all available models. Does this require to launch a full evaluation against all models?
> >
> > Computing the adversarial accuracy is cheap, compared to the cost of PubDef training, because it suffices to evaluate the accuracy on only a small random subset of the validation set. In the paper, we report results on the entire validation set, but in practice, it would suffice to use a subset of the validation set.
> >
> > ### Types of defenses
> >
> > > the authors state that they propose "using all available system-level defenses", however the presented method is clearly a ML-level defense. The authors should clarify this aspect, as it is confusing to read and there is no discussion in the rest of the paper on which system-level defenses are mounted on the model, or if they are mounted at all.
> >
> > We will revise the paper accordingly. We propose that PubDef's training method (an ML-level defense) be combined with all relevant systems-level defenses (as was stated in Section 1, "We propose using...") to protect against attackers under all threat models without a loss in the model utility. The specific systems-level defenses are listed in the last sentence of Section 2. See also assumption 3 in Section 3 and point 3 in Section 7.3.
> >
> > ### Clean accuracy of adversarial training
> >
> > > the authors state that the drawback of AT is that it degrades clean accuracy, however this does not seem to be the case https://robustbench.github.io. The authors should clarify this aspect by detailing this statement.
> >
> > Yes, adversarial training does degrade clean accuracy [1]. For instance, for ImageNet (with no additional training data), a ResNet-50 can achieve 79-80% clean accuracy. In comparison, the best adversarial training recipe for ResNet-50 achieves 63-64% clean accuracy [2]. Similarly, for CIFAR-10, undefended WRN-34-10 models achieve about 96% clean accuracy but adversarial training achieves about 85% clean accuracy [3] (when reviewing RobustBench, make sure to compare only to schemes with no extra training data or synthetic images for a fair comparison).
> >
> > Thank you for reading our rebuttal. Please don’t hesitate to let us know if there are other questions or concerns we have not addressed.

---

> > > ### Author Response · Authors · 2023-11-17
> > > **Author response to Reviewer nPyj (3/3)**
> > >
> > > References
> > >
> > > - [1] Tsipras et al., Robustness May Be at Odds with Accuracy, ICLR 2019.
> > > - [2] Salman et al., Do Adversarially Robust ImageNet Models Transfer Better?, NeurIPS 2020.
> > > - [3] Addepalli et al., Scaling Adversarial Training to Large Perturbation Bounds, ECCV 2022.

---

> > > > ### Comment · Reviewer_nPyj · 2023-11-20
> > > > **Official Comment by Reviewer nPyj**
> > > >
> > > > I acknowledge the authors' responses, and I thank them for their attention in clarifying their work.
> > > >
> > > > My points on the choice of public models, cost of training, clarification on the type of defenses, and clean accuracy of AT have been fully addressed.
> > > >
> > > > However, my crucial point on "it is not necessary in practice to make every model robust against white-box attacks" still stands and the fact that this defense does not improve even by a little the robustness against white box attacks is concerning. Why would one choose this method over AT, when AT provides robustness against black-box **and** white-box attacks? Furthermore, I expect the boundary to be more robust even with transfer attacks, but this is not the case. Thus, what is this model learning exactly?
> > > >
> > > > I also want to clarify that the ensemble attack that I requested is not the one that the authors show in their paper (where they take the best optimization result out of the pool of adversarial perturbations). The attack I wanted to see is one that uses for the optimization the mean gradient gathered from a set of models instead of separate runs of attacks. I apologize if that was not clear from my initial review.
> > > >
> > > > For these observations, I retain my original score.

---

> > > > > ### Author Response · Authors · 2023-11-21
> > > > > **Author response to Reviewer nPyj**
> > > > >
> > > > > Thank you for the prompt response! We wonder if there may be a misunderstanding. AT does not provide satisfactory robustness against black-box attacks, nor does it provide satisfactory clean accuracy in the absence of an attack. There are two reasons to use PubDef over AT:
> > > > > 1. **PubDef has much better clean accuracy than AT and is plausibly deployable** (AT has unacceptably bad clean accuracy; e.g., on ImageNet, an undefended model has 80% clean accuracy, AT has 63% clean accuracy, and PubDef has 78% clean accuracy; because of the drop in clean accuracy, AT is not deployable in many practical settings; and it doesn't matter how secure a defense theoretically is, if it can't be deployed), and
> > > > > 2. **PubDef has much better robust accuracy against black-box TAPM attacks than AT** (on ImageNet, AT achieves 36% robust accuracy against TAPM attacks, vs 63% for PubDef).
> > > > >
> > > > > See Table 1, and compare "Best white-box adv. train" to "PubDef (ours)".
> > > > >
> > > > > Regarding "more robust even with transfer attacks", PubDef's decision boundary is more robust than AT's decision boundary against TAPM transfer attacks. The purpose of our work is not to build classifiers whose decision boundary is more aligned with human perception, but rather to improve practical security against the attacks that would be easiest to conduct in the wild (namely, TAPM attacks).
> > > > >
> > > > > Regarding the ensemble, thank you for your persistence in raising this point and helping us understand your concern. We confirm that we have indeed conducted such an experiment. After reviewing the paper again, we realized that our writing was not clear, but we did average gradients in the ensemble attack reported in Section 7.1. We can see how the paper was confusing in its description of our ensemble attack and we apologize for the confusion. We constructed an ensemble of four source models (one source model randomly chosen from each group) and constructed a gradient attack based on the average of logits of these four models. We repeated this three times, constructing three ensembles (each ensemble with a different random choice of source models), and applied 11 attack algorithms to each ensemble, so for each sample, we generated 33 attack candidates (one per ensemble and attack algorithm), chose the best of these 33 candidates, and measured on the best-of-33 attack. Each of those 33 candidates was generated by averaging gradients from 4 models. We found that PubDef is robust against such an ensemble attack, and this ensemble attack was no better than an attack constructed from the best single-source model.
> > > > >
> > > > > Technical detail: our ensemble averaged the logits, rather than averaging the losses or softmax scores; in our past experience, we have found that all three choices yield similar performance, but we could certainly try other choices if required for acceptance. We will revise the paper to present our experiment more clearly. We would be glad to conduct and report on additional ensemble attacks if required for acceptance.
> > > > >
> > > > > Please let us know if there is still any other clarification we can provide. We will revise the original text in the paper to address these points and are glad to receive any further guidance on how we can improve the writing.

---

### Official Review · Reviewer_DtJG · 2023-10-31

**Soundness:** 4 excellent
**Presentation:** 4 excellent
**Contribution:** 2 fair
**Rating:** 6
**Confidence:** 3

**Summary:**

The authors approach transfer attacks from the practical perspective and propose training procedure that allows to achieve robustness with a small drop in clean accuracy. They empirically show that their rather simple approach to selecting models allows good performance with respect to SOTA white-box defences. They do this by performing extensive evaluations on a wide range of public models

**Strengths:**

-	The paper considers a real-life scenario of defending specifically against black-box attacks.
-	Evaluation is very extensive (considering different 264 combinations of source models and attack mechanisms)
-	Solid results with potentially high relevance for real-life industrial applications

**Weaknesses:**

The authors state the weaknesses and limitations of their approach quite fully in Section 7.3. I could add that although proposed heuristics and empirical results could be significant in practical applications, the methodological contribution of this work is rather incremental.

**Questions:**

The authors state that PubDef achieves much better results than SOTA models from RobustBench. Where in the main paper can we find the clean and robust accuracy of these SOTA models to compare? I found the presentation of the results a bit confusing.

---

> ### Author Response · Authors · 2023-11-17
> **Author response to Reviewer DtJG**
>
> Thank you for taking the time to review our paper. We really appreciate your comments and constructive feedback. Please see the answers and clarifications to your questions below.
>
> ### Contributions
>
> > The authors state the weaknesses and limitations of their approach quite fully in Section 7.3. I could add that although proposed heuristics and empirical results could be significant in practical applications, the methodological contribution of this work is rather incremental.
>
> We would like to add that the practical contribution is enabled by a scientific discovery on the generalization ability of the defense against unseen attacks. We did not emphasize this in the paper because it is not our main contribution, but we believe that this is a surprising result that has not been documented before. We attempt to explain this phenomenon in Section 7.2 through some empirical measurements. We also give our intuition in Appendix A.5.3.
>
> ### SOTA models from RobustBench
>
> > The authors state that PubDef achieves much better results than SOTA models from RobustBench. Where in the main paper can we find the clean and robust accuracy of these SOTA models to compare? I found the presentation of the results a bit confusing.
>
> For SOTA models from RobustBench, please see Table 1 ("Best white-box adv. train") for the clean and robust accuracy of these models. We compare our defense to the most robust models on [RobustBench](https://robustbench.github.io/) (at the time of submission) with the same architecture as our PubDef models.
>
> For completeness, the white-box adversarially trained models we use (under “Best white-box adv. train” in all tables) are:
> - CIFAR-10, WideResNet-34-10: Addepalli et al., Scaling Adversarial Training to Large Perturbation Bounds, ECCV 2022.
> - CIFAR-100, WideResNet-34-10: Addepalli et al., Efficient and Effective Augmentation Strategy for Adversarial Training, NeurIPS 2022.
> - ImageNet, ResNet-50: Salman et al., Do Adversarially Robust ImageNet Models Transfer Better?, NeurIPS 2020.
>
> Thank you for reading our rebuttal. Please don’t hesitate to let us know if there are other questions or concerns we have not addressed.

---

> > ### Comment · Reviewer_DtJG · 2023-11-21
> > **Response**
> >
> > Thank you for your response. I am retaining my score.

---

> > > ### Author Response · Authors · 2023-11-22
> > >
> > > Thank you for taking the time to read and acknowledge our response! Please don’t hesitate to let us know if there is anything else we can clarify.

---

### Official Review · Reviewer_nHrU · 2023-11-05

**Soundness:** 2 fair
**Presentation:** 3 good
**Contribution:** 2 fair
**Rating:** 6
**Confidence:** 5

**Summary:**

The paper argues that the common modality for attack on ML models (image classifiers are used in experiments) is going to be via transfer attacks on public models since companies are likely to keep their model weights protected. The paper then recommends defenders to adversarially train against a reasonable subset of  public models and find that works against L-inf norm attacks that are transferred from public models. They call their technique PUBDEF.

The authors present the work in a game-theoretic perspective where the attacker only has access to public models.

The authors acknowledge some of the limitations of the work. In my judgement, they are quite significant. They mention that if an attacker somehow is able to infer model weights (say by training a surrogate model), then they can bypass the defense. For typical classifiers, I think this is a significant concern. One limitation that they do not appear to consider is the possibility of black-box attacks directly on the model (no transfer attack needed) and do not evaluate their defense against a black-box attacker. Thus, the setting is somewhat limited in which the attacker can only do transfer attacks on the protected model and nothing else.


Overall, the motivation for the paper, the defense approach used, and evaluation all need to be better.

**Strengths:**

The threat model of transfer attacks is well known. The authors assume that this is the main likely modality in practice. That assumption is likely a reasonable assumption only in very complex and large models. In most other cases, including datasets and models that the authors consider, there are other practical attack strategies, including blackbox attacks and model stealing attacks. To authors' credit, they acknowledge some of these limitations (especially model stealing attack via training a surrogate), but that doesn't make the assumption more realistic.

Given their assumption, their experimental results appear reasonable. The main contribution is that the defender can choose a small subset of public models against which to adversarially train their model. They require that public models be trained on the same task.  In practice, they find that such a model is often robust against a broad range of adversarial attacks that are restricted to using the public models.

**Weaknesses:**

Weaknesses are unfortunately significant.

-- The assumption underlying the paper is likely unrealistic in that most blackbox models will also permit querying. Thus, transfer attacks are not the only option for an adversary. Blackbox attacks are also a possibility and, in fact, may be the primary attack strategy. The proposed defenses were not evaluated against Square attack and other blackbox attack strategies.


[Rebuttal]

I considered the rebuttal provided so far.  The main concern its that the approach is susceptible to blackbox attacks (model stealing attacks are a potential issue as well, but I think the primary attack vector is likely to be blackbox attacks). An attacker will choose the best attack strategy available among many options, including transfer attacks, blackbox attacks, and model stealing attacks. The authors present preliminary results that showed that blackbox attacks would succeed and they would thus either need stateful defenses to work (which were unfortunately recently broken by Feng et al. 2023) or they need to combine their scheme with a noise-based blackbox defense where inputs are combined with noise before doing an inference. The latter combination showed some promise, exceeding the performance of a simple white box defense, but significantly lowering the reported natural accuracy and adversarial accuracy of PubDef by about 10% each in Table 1.

I am raising my score, assuming that the authors are willing to include a deeper analysis of their work against blackbox attacks on CIFAR-10 and CIFAR-100 in the final paper and include a clear acknowledgment something along the following lines in bullet 3:

Blackbox attacks are a potential concern against PubDef. When we started this work, a stateful defense was available, which would have thwarted blackbox attacks and could be combined with our scheme  with no or little change to reported results. Unfortunately, very recently,  Feng et al. broke current stateful defenses, but did not rule out potential new stateful defenses. If new stateful were to be developed, they should be combined with your scheme. In the absence of stateful defenses being available, we present preliminary findings  against a blackbox attacker (see Appendix, section X). They suggest that PubDef, unmodified, would be vulnerable to blackbox attacks and not provide an adequate defense. However, PubDef in combination with a noise-based defense shows some promise (see Appendix, Section X). For example, on the CIFAR-10 dataset, with a sigma of 0.02, PubDef  has an adversarial accuracy of 79.8 against the best attacker, with a natural accuracy of 88.9. This is a drop of adversarial accuracy  and natural accuracy of about 10% each  from the  results in Table 1 that only factored in transfer attacks, but still superior to simply using best white-box adversarial training in both clean and natural accuracy. [And then report additional results on other datasets and include them in the paper.]

**Questions:**

I recommend authors to consider a different setting for their work where transfer attacks are more common due to the complexity of stealing models or doing direct blackbox attacks. The setting would be attacks on generative AI models such as ChatGPT. Recent work shows that transfer attacks are possible on such models (see a 2023 paper by Zico Kolter, Matt Fredrikson and others that  was also mentioned in a recent NYTimes article) where the attacker is able to append a suffix to a prompt to cause the model to output something that can be considered harmful in some way. Unfortunately, though, adversarial training is not the defense approach that is used in such settings (at least until now). So, the approach presented by the authors in this paper is unlikely to work and may require significant changes. But, the setting is likely more realistic for the assumptions they rely on.

---

> ### Author Response · Authors · 2023-11-17
> **Author response to Reviewer nHrU (1/2)**
>
> Thank you for taking the time to review our paper. We really appreciate your comments and constructive feedback. Please see the answers and clarifications to your questions below.
>
> ### Black-box attacks
>
> > One limitation that they do not appear to consider is the possibility of black-box attacks directly on the model (no transfer attack needed) and do not evaluate their defense against a black-box attacker.
>
> We do consider black-box attacks. There are two types of black-box attacks: transfer attacks and query-based attacks. We propose using “the right tool for the job”: we propose defending against query-based attacks with system-level defenses, and defending against transfer attacks with PubDef's training method (an ML-based defense). As mentioned in the paper, there is an orthogonal line of research that develops system-level defenses against query-based attacks, such as stateful detection [1] and noise addition [2]. We do not attempt to innovate on such system-level defenses and recommend practitioners deploy such defenses in concert with PubDef.
>
> Cloud API (e.g., Clarifai, Google Vision AI, or Azure AI Vision) is an application where PubDef can be very helpful. First, the model weights are private by default; this already stops white-box attacks. Then, system-level defenses like [1] or [2] can be deployed to stop the query-based attacks. Finally, when combined with PubDef, we have a system that addresses all the attack threat models effectively and minimally sacrifices utility.
>
> Alternatively, in settings where the cost of a failed attack is extremely high, PubDef can be deployed without any system-level defenses against query-based attacks. This means the adversary wants to come up with the best possible attack before submitting it to the victim system. This rules out query-based attacks as they require submitting lots of adversarial inputs most of which are failed attacks. For instance, with supervised biometric authentication, a few failed attempts at fooling a biometric authentication may lead to the adversary being arrested, as the attack is immediately detected and the adversary must be present in person. Another example is social media bots. Creating a fake account is very costly; it often requires some degree of identification (e.g., phone number or email address) and connection to real users. A failed attack would lead to these accounts being permanently banned.
>
> - [1] Stateful Detection of Black-Box Adversarial Attacks, https://arxiv.org/abs/1907.05587
> - [2] Random Noise Defense Against Query-Based Black-Box Attacks, https://arxiv.org/abs/2104.11470
>
> ### Surrogate models
>
> We agree it is likely possible to collect a large dataset, label those samples manually, train a surrogate model, and use it in a transfer attack (see Section 7.3, item 2). However, such an attack would likely be very expensive in practice, due to the labeling and the training cost. It will additionally require expertise in machine learning as well as in the attacked domain. In security, greatly increasing the cost of attacks is often sufficient to deter attackers or render attacks unprofitable. Often, if an attacker has sufficient funding and motivation, it may not be possible to stop them.

---

> ### Author Response · Authors · 2023-11-17
> **Author response to Reviewer nHrU (2/2)**
>
> ### Simple defense strategy (white-box adversarial training)
>
> > Another limitation of the work is why the defender only has to use public models for adversarial training. It appears that the defender should be able to adversarially train against white box attack model (since the defender knows all the weights), even if the attacks are less powerful and are transfer attacks. A standard defense should have been evaluated.
>
> > No evaluation against a simpler defense strategy of simply adversarial training the model using the standard techniques (e.g., PGD), assuming a white box attacker and then evaluating that defense against transfer attacks.
>
> We appreciate the suggestion. We already evaluated it; the state-of-the-art white-box adversarially trained models lose to our method in both the clean accuracy and robustness against the transfer attacks. See Table 1 (“Best white-box adv. train”). We emphasize that we compare our defense to the most robust models (at the time of submission) on [RobustBench](https://robustbench.github.io/) that do not use extra or synthetic training data and have the same architecture as our PubDef models.
>
> Furthermore, deploying white-box adversarial training in practice is completely unrealistic; it incurs an unacceptably large penalty for clean accuracy. For instance, for ImageNet, this simple defense degrades clean accuracy from 80% to 63% (Salman et al., 2020). Our scheme completely avoids this problem.
>
> For completeness, the white-box adversarially trained models we use (under “Best white-box adv. train” in all tables) are:
> - CIFAR-10, WideResNet-34-10: Addepalli et al., Scaling Adversarial Training to Large Perturbation Bounds, ECCV 2022.
> - CIFAR-100, WideResNet-34-10: Addepalli et al., Efficient and Effective Augmentation Strategy for Adversarial Training, NeurIPS 2022.
> - ImageNet, ResNet-50: Salman et al., Do Adversarially Robust ImageNet Models Transfer Better?, NeurIPS 2020.
>
> ### GenAI
>
> > I recommend authors to consider a different setting for their work where transfer attacks are more common due to the complexity of stealing models or doing direct blackbox attacks. The setting would be attacks on generative AI models such as ChatGPT.
>
> This would be an exciting application of our methods, but unfortunately, LLMs are far beyond our computational resources to evaluate. Training an LLM with adversarial training is significantly more expensive than training an LLM from scratch. As far as we are aware, no group has yet reported successful adversarial training of a non-trivial LLM. Using PubDef on an LLM directly would also require generating transfer attacks on the training and the test sets. The only transfer attack that reliably works against LLMs (Zou et al., 2023) takes about an hour to generate an attack for a single sample on an Nvidia A100 GPU, so applying adversarial training is infeasible with current methods.
>
> > Unfortunately, though, adversarial training is not the defense approach that is used in such settings (at least until now). So, the approach presented by the authors in this paper is unlikely to work and may require significant changes. But, the setting is likely more realistic for the assumptions they rely on.
>
> We believe that the fact that white-box adversarial training does not work makes our approach even more attractive. As mentioned in Section 1, the two reasons white-box adversarial training is not used in practice are (1) the training cost and (2) the accuracy drop. Our method exactly deals with both of these issues; unlike adversarial training, it is more efficient to train, and it has a much better robust-accuracy trade-off.
>
> Thank you for reading our rebuttal. Please don’t hesitate to let us know if there are other questions or concerns we have not addressed.

---

> ### Comment · Reviewer_nHrU · 2023-11-21
> **Feedback to the authors on their response**
>
> Current stateful defenses are now unfortunately broken. See the following paper:
>
> https://arxiv.org/abs/2303.06280
>
> I think you cite this work already.  I think there is a reasonable doubt as a result of this work  if one can rely on system-based defenses against blackbox attacks. An adaptive attacker, that factors in the noise injection, may also be able to overcome the noise-based method you mention -- at least, there is good reason to not entirely rely on such defenses. They also will degrade accuracy in the clean case.
>
> So, I suggest disclosing the results against blackbox attacks as well, even if they make your defense look bad. That sets a potential bound on the effectiveness of the defense against a motivated adversary.  You can point out how the results would change if noise were to be injected in the queries and whether you can achieve close to reported adversarial accuracy with a reaonsable level of noise injection and not hurting clean accuracy significantly in that process.
>
> (My point in the above is that the issue of defending against a blackbox adversary is not entirely orthogonal. It impacts achievable clean accuracy and adversarial accuracy, given your assumptions.)
>
> Thanks for clarifying that you considered white box defenses in your comparison in Table 1. The evaluation there seems reasonable.

---

> > ### Author Response · Authors · 2023-11-22
> > **Author response to Reviewer nHrU**
> >
> > Thank you for your thoughtful evaluation! We share your concern. If systems-level defenses are ultimately ineffective against query-based attacks, then our approach will be of much more limited applicability (e.g., only to applications where the adversary is caught immediately if the attack fails). The Feng et al paper is a very significant result that should raise questions about our approach, but it is not clear to us where the cat-and-mouse game between query-based attacks and system-level defenses will ultimately land. Feng et al list two defenses that are not known to be broken (AAA and AdvMind), so it might be premature to conclude that all defenses against query-based attacks are broken. As we write in the submission (Section 7.3), “It is not yet clear whether known defenses will be effective against a knowledgeable attacker.”
> >
> > We evaluated the models against the Square attack (score-based) [1], and PubDef has better accuracy than an undefended model against the Square attack, but worse than adversarial training. For ImageNet and 100-step Square attack, undefended model: 50% adversarial accuracy, white-box AT: 58%, PubDef: 55%. Of course, white-box AT has an unacceptable impact on clean accuracy, which PubDef does not suffer from. For results under the 1000-step Square attack and on all the datasets, see Table A below.
> >
> > With more queries, the adversarial accuracy of both the undefended and the PubDef models reduces to zero as they are not white-box robust. Regardless, the PubDef models are always more robust than the undefended and can provide substantial protection against the Square attack under some settings. We will include this result in the paper, as you have suggested.
> >
> > **Table A**: Adversarial accuracy under Square attack (score-based) with 100 / 1000 queries.
> >
> > | Models            |    CIFAR-10 |   CIFAR-100 |    ImageNet |
> > |-------------------|------------:|------------:|------------:|
> > | No Defense        | 36.6 /  0.2 |  9.1 /  1.8 | 49.9 / 17.9 |
> > | Best White-Box AT | 77.9 / 67.3 | 55.9 / 41.0 | 58.4 / 57.6 |
> > | PubDef            | 55.2 /  8.8 | 13.5 /  0.3 | 55.1 / 32.3 |
> >
> > We also conducted additional experiments under hard-label or decision-based attacks (see Table B below). Note that unlike the Square attack, these attacks are generally “distance-minimizing” rather than “loss-maximizing.” With the HopSkipJump attack [2], the trend on CIFAR-10 and CIFAR-100 is similar to the Square attack above. However, on ImageNet, the result is rather unexpected with the model without defense coming out on top in terms of robustness. So we ran another hard-label different attack, GeoDA [3], which now shows white-box AT as the most robust (mean $\ell_\infty$-norm of 0.1662) followed very closely by PubDef (0.1604).
> >
> > **Table B**: Mean adversarial distance ($\ell_\infty$-norm) under decision-based attacks (hard labels) with 1000 queries. The largest distance (i.e., most robust) is in bold. The attack success rates are not 100%, but all are roughly the same (around 90%) so we omitted them from the table.
> >
> > | Models            | CIFAR-10 (HSJ) | CIFAR-100 (HSJ) | ImageNet (HSJ) | ImageNet (GeoDA) |
> > |-------------------|---------------:|----------------:|---------------:|-----------------:|
> > | No Defense        |         0.0276 |          0.0208 |     **0.1995** |           0.1186 |
> > | Best White-Box AT |     **0.1542** |      **0.1518** |         0.1969 |       **0.1662** |
> > | PubDef            |         0.0466 |          0.0397 |         0.1814 |           0.1604 |
> >
> > We will be sure to include these results in the revised paper. We use the official implementation from https://github.com/Trusted-AI/adversarial-robustness-toolbox and all default attack parameters.
> >
> > - [1] [1912.00049] Square Attack: a query-efficient black-box adversarial attack via random search
> > - [2] [1904.02144] HopSkipJumpAttack: A Query-Efficient Decision-Based Attack
> > - [3] [2003.06468] GeoDA: a geometric framework for black-box adversarial attacks

---

> > > ### Comment · Reviewer_nHrU · 2023-11-23
> > >
> > > Any results on noise-based blackbox defenses? How well do they combine with your defense?

---

> > > > ### Author Response · Authors · 2023-11-23
> > > >
> > > > Thank you for the suggestion! We ran an experiment using the defense proposed by Qin et al. [1] which adds a Gaussian noise to the input image.
> > > >
> > > > **Setup.** We evaluated the defense when combined with three models (the standard, the white-box AT, and the PubDef models) as in all the previous experiments. We chose the standard deviation of the additive noise ($\sigma$) as 0.01 and 0.02 and followed the same evaluation procedure as Qin et al. We evaluated the defense against the Square attack with 100 and 1000 queries.
> > > >
> > > > **Results.** The PubDef models have both higher clean accuracy (4–7 percentage points) and higher adversarial accuracy (~1 pp.) than the white-box AT models. Adding PubDef to the Qin et al. defense boosts the robustness up to 24 pp. (from 66 to 80) against the 1000-query Square attack.
> > > >
> > > > **Table C**: Accuracy on the CIFAR-10 dataset under no attack (i.e., clean accuracy) and Square attack with 100 queries and 1000 queries.
> > > >
> > > > | Models                            | No Attack | Square-100 | Square-1000 |
> > > > |-----------------------------------|----------:|-----------:|------------:|
> > > > | No Defense ($\sigma=0.01$)        |      92.0 |       71.0 |        53.2 |
> > > > | No Defense ($\sigma=0.02$)        |      89.2 |       75.4 |        66.2 |
> > > > | Best White-Box AT ($\sigma=0.01$) |      85.2 |       80.7 |        76.5 |
> > > > | Best White-Box AT ($\sigma=0.02$) |      85.0 |       81.7 |        78.9 |
> > > > | PubDef ($\sigma=0.01$)            |  **92.6** |       81.4 |        75.6 |
> > > > | PubDef ($\sigma=0.02$)            |      88.9 |   **82.1** |    **79.8** |
> > > >
> > > > If we are able to submit more responses after the deadline, we will try to add more results on the other datasets and on the decision-based attack.
> > > >
> > > > [1] Qin et al., Random Noise Defense Against Query-Based Black-Box Attacks, NeurIPS 2021. https://arxiv.org/abs/2104.11470

---

### Meta-Review · Area_Chair_3AgC · 2023-12-07

**Metareview:**

"Defending Against Transfer Attacks From Public Models" discusses adversarial attacks against models behind an API. The submission argues that 'transfer from public models' is a missing ingredient in model robustness, and proposes a strategy to safeguard models against such transfer attacks. Evaluating their approach, the authors find it to be better than adversarial training when attackers are constrained to API access only..

Reviewers broadly support the submission, but a big point of uncertainty and concern is whether the 'transfer from public models' really is the missing piece in adversarial robustness. These questions focus on the framing of this work, not necessarily on the concrete approach proposed to defend against transfer from public models.
Reviewers note, for example, that query-based black-box attacks are also possible (and several new query attacks are brought up that should be discussed in the updated submission), which undermine the impact of defenses against transfer from public models only. There were also concerns regarding the strength of the current evaluation, and whether stronger, adaptive attacks against transfer from public models are possible.

Ultimately, I think this is clearly an interesting submission, and I encourage all participants to continue the discussion regarding the framing of this work, and the validity of the slogan put forth in the intro, at the poster session.

I do require the authors to include all material provided during the author response in the camera-ready version, especially the additional data regarding unseen public models, the discussion of novel query-based attacks with reviewer nHrU and the clarification of the ensemble attack as discussed with reviewer nPyj.

**Justification For Why Not Higher Score:**

Concerns regarding the framing of the work and uncertainty regarding the strength of the evaluation.

**Justification For Why Not Lower Score:**

N/A

---

### Decision · Program_Chairs · 2024-01-16

Accept (poster)